# Exploring Modality Collaboration with Modality-Agnostic Transformers in Multi-Modal Federated Learning

## Abstract

In Federated Learning (FL), the focus has predominantly been on uni-modal scenarios, limiting the system's ability to leverage multi-modal data. This paper introduces a novel setting, Modality-Collaborated Federated Learning (MCFL), designed to facilitate collaboration among uni-modal clients with different data modalities. Unlike existing frameworks that emphasize multi-modal clients and tasks, MCFL aims to be more practical by focusing on uni-modal clients and ensuring performance gains across individual modalities. To address the challenges of model heterogeneity and modality gaps in MCFL, we propose Federated Modality Collaboration (FedCola), a framework based on a modality-agnostic transformer. FedCola explores optimal strategies in cross-modal parameter-sharing, model aggregation, and temporal modality arrangement. Our comprehensive evaluations demonstrate that FedCola significantly outperforms existing solutions, serving as a robust baseline for MCFL and marking a substantial advancement in federated learning.

## 1 Introduction

In the burgeoning field of Federated Learning (FL), collaborative and privacy-preserving learning can be achieved without the need to exchange raw data between participants (McMahan et al., 2017). In this framework, a central server cyclically disseminates a *global model* to selected clients for local training. The server then receives the trained models from these clients, aggregates them into an updated global model, and repeats this process over multiple rounds. In general, introducing more training data from more clients can benefit the performance of the learned global model (Mansour et al., 2020). Current FL methods primarily focus on uni-modal scenarios, where all participants contribute data of the same modality. Therefore, only the data on the clients with the same modality can contribute to the FL process. However, there are situations where some clients who gather data from a different modality are not eligible to participate in the FL system, which nonetheless share substantial high-level common knowledge. For instance, clients with medical transcription data cannot participate in an FL system for medical images. To better leverage the shared knowledge, studies that extend uni-modal FL systems into the multi-modal FL realm are desired.

There have been a few studies investigating multi-modal federated learning systems (Yu et al., 2023; Xiong et al., 2022; Feng et al., 2023). Their settings, however, aim to get better encoders for *multi-modal tasks*, such as image captioning and image-text retrieval, instead of facilitating the knowledge-sharing between different modalities. Those multi-modal tasks usually necessitate an explicit alignment with the *multi-modal data* annotated with multiple modalities. However, the privacy-preserving philosophy of federated learning prohibits sharing the raw format or features of the client data, meaning that such an alignment can only be performed inside the client itself. Consequently, *multi-modal clients* with multi-modal data serve as a secure hub to perform such an explicit alignment, making them indispensable under those settings. For instance, Xiong et al. (2022) introduced a unified framework that facilitates multi-modal learning within multi-modal clients by learning a fused multi-modal feature. This approach mandates that all participating clients be multi-modal, thereby excluding uni-modal clients from participating in the FL process. Beyond that, Yu et al. (2023) relaxes the requirements of the participating clients to further enhance the multi-modal performance by allowing uni-modal clients to participate with the help of a public dataset. Despite

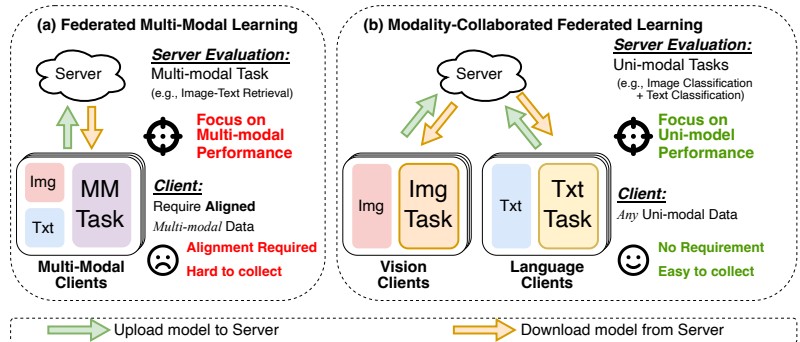

Figure 1: Previous federated multi-modal learning vs. our modality-collaborated federated learning. (a) Federated multi-modal learning focuses on multi-modal performance. The multi-modal knowledge is obtained from *the multi-modal task on multi-modal clients*. (b) Our proposed setting focuses on the collaboration between uni-modal clients to obtain performance gains for all modalities. The multi-modal knowledge is obtained from *aggregating parameters from different modalities*.

this inclusivity of uni-modal clients, the reliance on multi-modal clients still exists. We note this line of study as Federated Multi-Modal Learning (FMML) *with a focus on the multi-modal tasks and clients*, as illustrated in Figure 1(a).

However, there is a gap between the FMML setting and the real application. While FMML emphasizes multi-modal clients, collecting multi-modal data is often less feasible than assembling uni-modal datasets. Additionally, FMML assumes perfect alignment of multi-modal labels, a condition that becomes increasingly challenging as the number of modalities grows. In previous settings, evaluations have focused solely on multi-modal performance, which cannot explicitly show if each modality will benefit from federated learning. For instance, an encoder that performs well in image-text retrieval may not necessarily excel in image classification. Therefore, to make the setting more practical and appealing to uni-modal participants, it is crucial to report the performance for each individual modality.

**Motivation of a new FL setting:** Given the reliance on multi-modal data and multi-modal-focused evaluation, we seek a new setting where *uni-modal clients are the main concern*. Specifically, we aim to extend multi-modal learning beyond the confines of multi-modal clients, enabling collaboration among uni-modal clients while ensuring performance improvements across all modalities. To satisfy such needs, we introduce a novel setting, termed **Modality-Collaborated Federated Learning (MCFL)**. This approach shifts the focus towards fostering collaboration between uni-modal clients rather than pursuing modality alignment within multi-modal clients, as illustrated in Figure 1(b). In this proposed setting, we adhere to two primary principles: 1) each client is restricted to data from a single modality, and 2) the ultimate performance metric is derived from the individual performances across each modality. This setting is particularly relevant in scenarios where clients possess uni-modal data of different modalities. For instance, in a healthcare scenario where different hospitals hold different uni-modal data (e.g., X-rays, medical transcriptions, magnetic resonance imaging), these data from different patients are less likely to be aligned across clients, making traditional collaboration unfeasible. However, MCFL offers a viable alternative for such cases.

In MCFL, more challenges arise beyond previous settings. One of the challenges in MCFL lies in multi-modal aggregation. In traditional uni-modal federated learning, all clients share identical model architectures, and the global model is aggregated through a weighted average (McMahan et al., 2017). However, *model heterogeneity* is inherent for a multi-modal FL system, given that different modalities usually depend on distinct architectures, such as Convolutional Neural Networks (CNN) (He et al., 2016) for images and Long Short-Term Memory (LSTM) (Hochreiter & Schmidhuber, 1997) or transformers (Vaswani et al., 2017) for texts. Encouragingly, recent advancements in model architectures have shown the feasibility of using a modality-agnostic structure, such as the transformer, for effective encoding of multi-modal data (Bao et al., 2022; Chen et al., 2020; Gan et al., 2020; Kim et al., 2021; Li et al., 2021). This progress enables us to propose a framework with a modality-agnostic transformer for MCFL, effectively addressing the model heterogeneity problem.

However, although the modality-agnostic transformer can address the model heterogeneity challenge, the *modality gap* due to the unique characteristics of each modality is another significant challenge in MCFL. As shown in Section 4, we empirically find that simply applying a transformer will make the global model biased towards one modality, leading to catastrophic performance on the other modality. Therefore, exploring new FL strategies is urgent to address the modality gap

and propose a feasible MCFL framework. Targeting significant challenges in FL, we pose several research questions shown in Section 5, including parameter-sharing, aggregation, and round-specific operations. Based on the insights that answer the research questions, we propose a new framework called **Fed**erated Modality **Colla**boration (**FedCola**), based on the modality-agnostic transformer to enable better modality collaboration in MCFL.

The main contributions of this paper are as follows:

- *New Setting:* We introduce a novel setting in federated learning (*i.e.*, MCFL) that emphasizes modality collaboration between uni-modal clients to benefit all modalities, eliminating the reliance for multi-modal data on clients.
- *New Framework:* We systemically study the feasibility of modality-agnostic transformers to address the model heterogeneity and modality gap in MCFL and propose a new framework, FedCola.
- *Better Performance:* We adapt a widely-used uni-modal federated learning algorithm (FedAVG (McMahan et al., 2017)) and the state-of-the-art algorithm for FMML (CreamFL (Yu et al., 2023)) into MCFL. FedCola succeeds in outperforming them in comprehensive scenarios, showing superior performance as a baseline framework for MCFL.
- *New outlook:* We provide insight that multi-modal knowledge can also be obtained by aggregating the parameters from uni-modal data, not only from the aligned multi-modal data, which enlightens further work in multi-modal federated learning.

## 2 PROBLEM DEFINITION

Consider a heterogeneous FL system for classification tasks with $N$ clients in total. Each client (indexed with $i$) has its private dataset $\mathcal{D}_i$ with a specific modality $m_i \in \mathcal{M}$ (*e.g.*, Image, Text, or Audio). The set of clients with modality $m$ indices is denoted as $N_m$. The clients, along with a server, collaboratively train a multi-modal global model $\phi = \bigcup_m \phi^{(m)} \cup \phi^{(m_s)}$, where $\phi^{(m)}$ are modality-specific parameters for modality $m$ and $\phi^{(m_s)}$ are *shared parameters* across all modalities. The federated learning will last for $T$ rounds. In each round, each client $i$ downloads the modality-related parameters $\phi^{(m)} \cup \phi^{(m_s)}$ from the server, performs local training with its own private dataset, and sends it back to the server. Considering the clients will not always be online in practice, we assume the server can receive the local models from $K = rN$ clients in each round, where $r$ is the ratio of available clients. The server will further aggregate the received models $\phi_i^{(m)} \cup \phi_i^{(m_s)}$ to update the global models. Due to the model heterogeneity, *aggregation is only feasible between clients with the same modality*. The objective of this system is to minimize the empirical risk in all the modalities, which can be formulated as

$$\min_{\phi} \sum_{m=1}^{M} \mathcal{L}(\phi(\boldsymbol{X}^{(m)}), \boldsymbol{Y}^{(m)}) \tag{1}$$

where $\mathcal{L}$ is the loss function and $(\boldsymbol{X}^{(m)}, \boldsymbol{Y}^{(m)})$ are all client training data with a modality of $m$. *Note that neither the input nor the label space is required to be shared across modalities.* The server will hold an exclusive test set on each modality for evaluation. The final performance will be evaluated as the **equal-weighted arithmetic mean** of **Top-1 Accuracy among all modalities**. As vision and language are the most popular modalities in current studies, we focus on these two modalities for demonstration.

Considering the key processes in MCFL, we will describe a method with three different perspectives of view in the following sections: 1) **Parameter-sharing:** The shared parameters for all modalities, *i.e.*, $\phi^{(m_s)}$. 2) **Aggregation:** The server operations that aggregate the client models $\phi_i^{(m)}$ to the global model $\phi^{(m)}$ for each $m$. 3) **Temporal Modality Arrangement:** During the $T$ rounds, it is not mandatory to involve all modalities. Thus, participating modalities in each round can be arranged to balance the *focus* among all modalities. We term the round-specific arrangement for participating modalities as the *Temporal Modality Arrangement*.

## 3 A UNI-MODAL BASELINE: UNI-FEDAVG

Before discovering frameworks that build a positive collaboration between modalities, we need to establish a baseline involving *no collaboration* as a reference. Therefore, we adapt the uni-modal

federated learning method to MCFL as a baseline by separating the multi-modal federated learning system into several uni-modal ones. We cluster the clients with the same modality and aggregate them into a uni-modal model. Finally, we will get separate uni-modal models for each modality, as shown in Figure 2(a). This indicates the performance *without any modality collaboration*. Therefore, no parameters are shared across different modalities, and the server only aggregates models from the clients with the same modality. This strategy is denoted as the *intra-modality aggregation* for *all parameters*. Based on the FedAVG (McMahan et al., 2017) algorithm, it can be formulated as

$$\phi^{(m)} = \sum_{i=1}^{N_m} \frac{|\mathcal{D}_i|}{\sum_{j=1}^{N_m} |\mathcal{D}_j|} \phi_i^{(m)} \text{ for each } m. \tag{2}$$

We note this baseline as Uni-modal FedAVG (Uni-FedAVG). It can be described as: 1) Parameter-sharing: No parameter is shared across modalities. 2) Aggregation: Only intra-modality aggregation is applied. 3) Temporal Modality Arrangement: All modalities will participate in each round. We will use Uni-FedAVG as the baseline to evaluate the impacts of the modality collaboration.

## 4 PRELIMINARY PROTOTYPE: A MODALITY-AGNOSTIC TRANSFORMER

In contrast to Uni-FedAVG, which handles each modality separately, another strategy to mitigate the model heterogeneity is to unify the model for all modalities. A modality-agnostic transformer (MAT) can encode multi-modal data. Therefore, it enables cross-modal knowledge sharing. In general, a MAT can be decomposed into three parts: *embedding layers* to tokenize data in different modalities, *transformer blocks* to extract features, and the *head* for the task, which is illustrated in Figure 2(b). Since each modality requires distinct embedding layers to tokenize the input data and head to map the features to the modality-specific task, separate parameters for each modality are still required. Therefore, we will still perform the *intra-modality aggregation* on the embedding layers and the heads. The transformer blocks are *shared parameters* $\phi^{(m_s)}$ to learn multi-modal features. Consequently, the aggregation can be extended to all clients as the *inter-modality aggregation*. Formally, the inter-modality aggregation can be formulated as

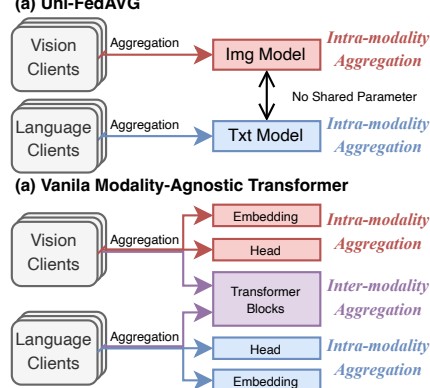

Figure 2: An example to encode multi-modal data with (a) separate uni-modal models and (b) a modality-agnostic transformer (MAT). Uni-FedAVG aggregates all parameters with *Intra-modality Aggregation*, while MAT aggregates transformer blocks with *Inter-modality Aggregation*.

$$\phi^{(m_s)} = \sum_{i=1}^{N} \frac{|\mathcal{D}_i|}{\sum_{j=1}^{N} |\mathcal{D}_j|} \phi_i^{(m_s)}. \tag{3}$$

We note this preliminary prototype as Vanilla MAT. Specifically, it can be described as 1) Parameter-sharing: The parameters of the *transformer blocks* are shared across modalities. 2) Aggregation: Intra-modality aggregation is applied for the embedding layers and heads, and inter-modality aggregation is applied for the transformer blocks. 3) Temporal Modality Arrangement: All modalities will participate in each round.

### 4.1 EVALUATION

To assess the efficacy of the vanilla MAT in MCFL, we conducted an empirical study utilizing a widely recognized dataset combination previously employed in multi-modal federated learning research (Yu et al., 2023), namely CIFAR-100 (Krizhevsky et al., 2009) and AGNEWS (Zhang et al., 2015). Detailed information regarding this study can be found in Appendix B.

Unfortunately, our findings reveal that the vanilla MAT is inadequate for effectively leveraging cross-modal information to improve performance across all modalities compared to Uni-FedAVG. As illustrated in Figure 3, the global model demonstrates a significant bias towards the language modality. This bias leads to a noticeable decline in performance for the vision modality (3.58%), although there is a slight improvement on language from 88.13% to 88.40%. It is worth noting

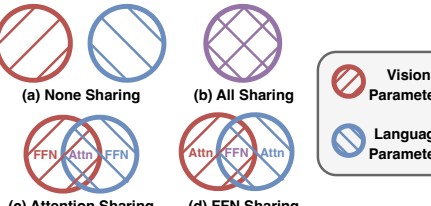

(a) None Sharing    (b) All Sharing

(c) Attention Sharing    (d) FFN Sharing

⬤ Vision Parameters

⬤ Language Parameters

Figure 4: Parameter-sharing strategies.

Table 1: Performance of each parameter-sharing strategy.

| Strategy | Img Acc | Txt Acc | Avg Acc |
|---|---|---|---|
| None Sharing (Uni-FedAVG) | 51.39 | 88.13 | 69.76 |
| All Sharing (Vanilla MAT) | 3.58 | 88.40 | 45.99 |
| Attention Sharing | **56.17** | **89.67** | **72.92** |
| FFN Sharing | 42.49 | 86.96 | 64.73 |
| Vision Attention Only | 52.52 | 88.57 | 70.55 |
| Language Attention Only | 25.32 | 87.83 | 56.58 |

that the total number of training samples in the language modality (120, 000) far exceeds that in the vision modality (50, 000). This imbalance causes the default inter-modality aggregation to weigh the parameters based on the number of samples, thereby overwhelming the contributions from the vision clients and resulting in a severe degradation in performance. Nonetheless, even balanced aggregation cannot mitigate such bias (Appendix C).

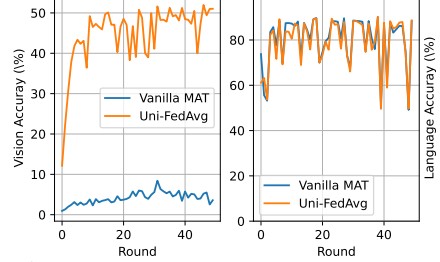

Figure 3: Performane of Vanilla MAT. It cannot benefit all modalities.

This observation underscores that merely deploying a multi-modal model in MCFL does not yield the anticipated benefits across all modalities, revealing a problematic lack of effective modality collaboration. Consequently, further research is essential to fully realize the potential of MAT in MCFL, extending beyond the limitations of the current preliminary prototype.

## 5 TOWARDS BETTER MODALITY COLLABORATION

Since the preliminary prototype shows unacceptable performance, in pursuit of the feasibility of MAT in MCFL, we pose several research questions aimed at optimizing the current preliminary prototype from different perspectives:

- **Parameter Sharing:** The current approach of sharing all parameters of the transformer blocks overlooks the capture of modality-specific knowledge to maintain the performance on each modality. *Which transformer block parameter should be shared across modalities? (**RQ1**)*
- **Aggregation:** The simple inter- and intra-modality aggregation tends to be biased. *Can better aggregation be achieved? (**RQ2**)*
- **Temporal Modality Arrangement:** All modalities currently participate in each round. Previous centralized learning works indicate that uni-modal pre-training before multi-modal learning can improve performance (Bao et al., 2022; Kim et al., 2021). *Can such a strategy lead to a better temporal modality arrangement for MCFL? (**RQ3**)*

Guided by our proposed research questions, we undertake empirical studies to scrutinize various alternatives and identify the most effective solution for each question. The experimental framework adheres to the setting established in Section 4.1.

### 5.1 RESEARCH QUESTION 1: CROSS-MODAL PARAMETER-SHARING

Given that sharing the *entire* transformer blocks can undermine the retention of modality-specific knowledge, it becomes imperative to devise a more granular parameter-sharing strategy that delves into the individual components of the transformer blocks. Beyond the previously discussed strategies of sharing all parameters (as Vanilla MAT) or sharing none (as Uni-FedAVG), we explore additional strategies informed by insights from prior research, as shown in Figure 4:

**Attention Sharing:** The self-attention layers stand as a pivotal element in a transformer block (Li et al., 2022). Leveraging the approach adopted from VLMo (Bao et al., 2022) in centralized settings, we consider the sharing of self-attention layers as a viable strategy to facilitate the capture of multi-modal features in MCFL.

**Feed-Forward Network (FFN) Sharing:** Contrasting with Attention Sharing, this approach entails sharing the feed-forward network to map the representations derived from each attention layer to the final features, which is also a feasible strategy in previous centralized study (Sung et al., 2023).

Our evaluations, detailed in Table 1, reveal that among the strategies tested, *Attention Sharing* emerges as the superior choice, showcasing the self-attention layers' adeptness at harnessing cross-

modal knowledge. To further substantiate that the capability of the multi-modal self-attention layer is stronger than the uni-modal ones, we assessed two additional strategies: using the self-attention layers from the vision (**Vision Attention Only**) or from the language (**Language Attention Only**) for all modalities in each aggregation. As anticipated, neither strategy surpassed Attention Sharing in performance, highlighting the significance of multi-modal knowledge.

## 5.2 RESEARCH QUESTION 2: CROSS-MODAL AGGREGATION

Given the insight that *Attention Sharing* is the most effective strategy at the parameter level, we further proceed to the aggregation level to mitigate the bias between modalities. Surprisingly, *Attention Sharing* with intra- and inter-modality aggregation can already mitigate the biases due to sufficient modality-specific information maintained in the feed-forward networks, which improves the vision performance from 3.58% to 56.17% (Table 1).

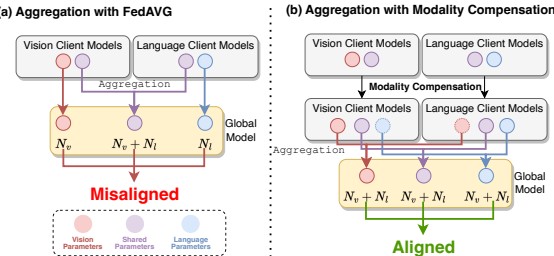

Figure 5: Alignment analysis of aggregation with (a) FedAVG and (b) our proposed Modality Compensation.

However, we note that aggregating shared and modality-specific layers may introduce potential *layer-level* misalignment for the final global model. As demonstrated in a previous generalization analysis (Mansour et al., 2020), a model's generalization error is bounded by the Rademacher complexity (Yin et al., 2019), with the empirical risk satisfying $\mathcal{O}\left(\frac{\sqrt{d_s + \log 1/\delta}}{|\mathcal{D}|}\right)$, where $d_s$ is the pseudo-dimension of the model, with a probability of at least $1 - \delta$.

Notably, the generalizability of the model is related to *the number of training samples*. Therefore, layers aggregated from different numbers of training samples may have *misaligned* generalizability. As the example given in Figure 5(a), when considering $N_v$ vision clients and $N_l$ language clients, the shared parameters are aggregated from $(N_v + N_l)$ clients, while the image-specific and language-specific parameters are aggregated from $N_v$ and $N_l$ clients respectively. This misalignment may damage the final performance of the global model.

To address this misalignment between shared and modality-specific parameters, we propose a *modality compensation* scheme to account for the reduced number of samples in intra-modality aggregation, which is shown in Figure 5(b). Specifically, before aggregation, we extend each client model to have *all* the parameters instead of those trained on the clients. The weights of the missing modality will be copied from the weights of the global model in the last round. A detailed algorithm is provided in Appendix F. In this way, we ensure all the layers of the global model are aggregated from the same number of clients and training samples, leading to a better-aligned global model. We further provide detailed demonstration in Appendix D.1 to show that aggregation with modality compensation will have the same *layer-level alignment* as applying FedAVG for all parameters.

## 5.3 RESEARCH QUESTION 3: TEMPORAL MODALITY ARRANGEMENT

In previous studies, we performed a straightforward temporal modality arrangement in which all modalities will participate in each round. However, existing multi-modal pre-training studies conducted in centralized settings (Kim et al., 2021; Bao et al., 2022) reveal that *initiating with uni-modal pre-training prior to multi-modal pre-training can enhance the final performance*. Such insights can be adapted to MCFL due to the *round-based structure* of federated learning. Therefore, we further adapt such training schemes in centralized settings to MCFL and evaluate the optimal strategy to enhance modality collaboration.

We term the adapted temporal modality arrangement strategy as *Modality Warm-up*. Since introducing more rounds for uni-modal pre-training will increase the total communication costs, we choose to split the *early rounds* to mimic such uni-modal pre-training instead of introducing more. We select one modality for pre-training and term it as the *Warming modality*, and the remaining modalities will be termed the *Warmed Modalities*. Similar to previous pre-training schemes, we will split all the communication rounds into three stages, delineated in Table 2 and described as follows: 1) **Warm-**

Table 2: Stages in Modality Warm-up. HD: Heat Distribution Stage. MM: Multi-Modal stage.

| Modality | Parameters | Stage | | |
|---|---|---|---|---|
| | | Warm-up | HD | MM |
| Warming | Modality-specific | Trained | Trained | Trained |
| | Shared | Trained | Trained | Trained |
| Warmed | Modality-specific | Not Participate | Trained | Trained |
| | Shared | Not Participate | Frozon | Trained |

Table 3: Performance of different modality warm-up strategies. HD means the Heat Distribution stage.

| Warming | HD? | CIFAR-100 + AGNEWS | | | OrganAMNIST + MTSamples | | |
|---|---|---|---|---|---|---|---|
| | | Img Acc | Txt Acc | Avg Acc | Img Acc | Txt Acc | Avg Acc |
| Vision | Yes | 55.23 | 88.13 | 71.68 | 92.60 | 35.64 | **64.12** |
| Vision | No | 57.26 | 90.20 | **73.73** | 93.36 | 22.31 | 57.84 |
| Language | Yes | 49.64 | 89.47 | 69.56 | 94.03 | 20.39 | 57.21 |
| Language | No | 51.11 | 89.61 | 70.36 | 92.32 | 21.54 | 56.93 |

**up Stage:** This initial stage is designed to *warm up* the shared parameters within clients utilizing the warming modality, thereby instilling prior knowledge. Since the server will only communicate with the clients with the warming modality, *the communication costs in this stage are significantly reduced*. 2) **Heat-Distribution Stage:** Following the warm-up phase, this *optional* stage aims to disseminate the *heat* or knowledge encapsulated in the shared parameters to the modality-specific parameters. Here, clients operating with the warmed-up modality will freeze the shared parameters and exclusively train the modality-specific parameters. 3) **Multi-Modal Stage:** In this final stage, a standard MCFL continues, where all parameters are participating, trained, and aggregated.

We studied four different warm-up strategies shown in Table 2. To better study the strategy with different modality correlations, we added one more medical scenario with OrganAMNIST (Yang et al., 2023) and MTSamples (South et al., 2012), which have a higher semantics correlation on the medical domain. As shown in Table 3, warming up on vision achieves better performance than on language in general, similar to the results in Section 5.1 that the Vision Attention Only yields better performance than Language Attention Only. It shows that the vision knowledge provides better initialization for the shared parameters during the warm-up, which aligns the insights under centralized training. However, the heat distribution stage doesn't always benefit the modality collaboration. It further improves the performance when there is a higher correlation between the modalities, i.e., OrganAMNIST and MTSamples. It indicates that the heat distribution stage helps to learn semantics-level instead of more general shared knowledge between modalities.

## 5.4 Put the Puzzles Together – Proposed Framework: FedCola

By exploring our research questions, we gain the following **remarks**: 1) Sharing the *self-attention layers* is the most effective parameter-sharing strategy. 2) Layer-wise misalignment exists in the vanilla aggregation strategy, which can be fixed by proposed *modality compensation*. 3) Warming up with *one specific modality* can provide a better initialization, thus improving the final performance.

With all the insights, we finally propose our FedCola framework, based on a modality-agnostic transformer and enhanced with Attention Sharing, Modality Compensation, and Modality Warm-up. It serves as a new baseline for MCFL. A detailed algorithm is provided in Appendix F. Although such insights are drawn from two-modal settings, our framework can be directly extended to scenarios with more modalities. We regard further studies with more modalities as our future work.

## 6 Experiments

To compare FedCola with previous methods that can be adapted to modality-collaborated federated learning, we conduct a comprehensive evaluation with challenging tasks.

### 6.1 Experimental Settings

**Datasets.** We consider two practical scenarios for evaluation: 1) Low-correlation datasets for general classification tasks: CIFAR-100 for vision and AGNEWS for language. It is a challenging setting to verify if *modality collaboration can still be achieved by general knowledge when semantics have a low correlation*. 2) Domain-specific medical datasets: OraganAMINIST for medical imaging data and MTSamples for medical transcripts data. It evaluates if *the shared high-level domain knowledge can be shared across modalities*.

**Federated Learning Settings.** We choose two federated learning settings based on common application scenarios: 1) Settings simulate **FL operated by several big parties** (*e.g.*, hospitals) where data is partitioned into fewer clients ($N_v = N_l = 4$), and 2) Settings simulate **FL operated by more devices**, where data is partitioned into more clients ($N_v = N_l = 16$). For each group of settings, we further choose three different data situations: 1) A **default** setting where data is partitioned into clients following a Dirichlet distribution with $\alpha = 0.5$ and clients will be sampled with $r = 0.5$, 2) A setting with **more data heterogeneity** where $\alpha$ drops to 0.1, and 3) A setting with **less stable**

Table 4: Evaluation under different FL scenarios. The bold style indicates the results in Uni-modal Accuracy (Img or Txt Acc) and Averaged Accuracy of all modalities (Avg Acc).

| Client # | Data Status | Method | CIFAR-100 + AGNEWS | | | OrganAMNIST + MTSamples | | |
|---|---|---|---|---|---|---|---|---|
| | | | Img Acc (%) | Txt Acc (%) | Avg Acc (%) | Img Acc (%) | Txt Acc (%) | Avg Acc (%) |
| $N_v = 4$ $N_l = 4$ | $\alpha = 0.5$ $r = 0.5$ | Uni-FedAVG | 51.39 | 88.13 | 69.76 | 91.89 | 20.64 | 56.27 |
| | | CreamFL | 51.74 | 90.04 | 70.89 | 90.69 | 20.39 | 55.54 |
| | | **FedCola (Ours)** | **57.26** | **90.20** | **73.73** | **92.60** | **35.64** | **64.12** |
| | $\alpha = 0.1$ $r = 0.5$ | Uni-FedAVG | 44.51 | 25.41 | 34.96 | 70.25 | 32.44 | 51.34 |
| | | CreamFL | 44.56 | 39.45 | 42.00 | 60.54 | 33.33 | 61.21 |
| | | **FedCola (Ours)** | **49.09** | **56.71** | **52.89** | **86.00** | **36.41** | **61.21** |
| | $\alpha = 0.5$ $r = 0.25$ | Uni-FedAVG | 41.97 | 66.93 | 54.45 | 89.69 | 30.26 | 59.97 |
| | | CreamFL | 41.65 | 66.67 | 54.16 | 85.09 | 32.05 | 58.57 |
| | | **FedCola (Ours)** | **51.51** | **79.53** | **65.53** | **91.68** | **33.97** | **62.83** |
| $N_v = 16$ $N_l = 16$ | $\alpha = 0.5$ $r = 0.5$ | Uni-FedAVG | 50.67 | **90.61** | 70.64 | 91.46 | 27.31 | 59.38 |
| | | CreamFL | 49.11 | 90.33 | 69.72 | 90.28 | 27.05 | 58.67 |
| | | **FedCola (Ours)** | **54.85** | 90.29 | **72.57** | **92.92** | **30.39** | **61.65** |
| | $\alpha = 0.1$ $r = 0.5$ | Uni-FedAVG | 45.76 | 49.05 | 47.41 | 80.30 | 35.13 | 57.72 |
| | | CreamFL | 44.93 | **52.49** | 48.71 | 85.26 | 33.46 | 59.36 |
| | | **FedCola (Ours)** | **49.04** | 49.29 | **49.17** | **87.86** | **35.26** | **61.56** |
| | $\alpha = 0.5$ $r = 0.25$ | Uni-FedAVG | 48.68 | 89.33 | 69.01 | 90.82 | 33.21 | 62.02 |
| | | CreamFL | 46.07 | 89.20 | 67.64 | 90.95 | 34.10 | 62.53 |
| | | **FedCola (Ours)** | **50.73** | **90.62** | **70.68** | **92.76** | **35.90** | **64.33** |

**clients** where $r$ drops to 0.25. The FL will last for 50 communication rounds, and there are $K = 4$ clients online in each round (the ratio $r = 0.5$). Each online client will receive and train the global model for 5 epochs with its local data and send it back to the server.

**Model Architecture.** We employ a pre-trained ViT-Small (Dosovitskiy et al., 2021) as the transformer blocks. Besides, images are embedded with a patch embedding layer with a patch size of 16, and texts are embedded with a BERT tokenizer (Devlin et al., 2018). The final prediction for each modality will be given by a corresponding classification head.

**Comparison Methods.** We provide several solutions from previous work as comparison methods. 1) Training Separate uni-modal models with FedAVG (McMahan et al., 2017) (Uni-FedAVG), which is the most wide-used algorithm in conventional FL. 2) CreamFL (Yu et al., 2023), which is the state-of-the-art KD-based method for modality-aligned federated learning. We adapt it to the MCFL with MS-COCO (Lin et al., 2015) as the public dataset, which follows their original design. *To maintain a fair comparison, all the methods use the same model architecture.* We discussed more related work and their relationship with MCFL in Appendix A and more implementation details in Appendix G.

**Evaluation Metrics.** We will report a separate and averaged **Top-1 accuracy** for **each** modality.

## 6.2 PERFORMANCE ANALYSIS

Table 4 shows the results under different FL scenarios. In general, FedCola outperforms all the comparison methods in all averaged and almost all uni-modal accuracies. Compared to Uni-FedAVG, FedCola accomplishes notable improvement for each modality. FedCola learns a shared attention mechanism to benefit the performance on both involved modalities. It reveals that FedCola can leverage more training samples that are not limited to the data in the same modality, therefore significantly improving the performance of the modality *with insufficient data*. Compared to CreamFL, FedCola outperforms CreamFL for up to $8.58\%$ in average accuracy with *no additional computation and communication costs*. We further analyze the resource requirements for each method in Section 6.3. Meanwhile, with the absence of multi-modal clients for direct feature alignment, CreamFL cannot always outperform Uni-FedAVG, demonstrating the difficulty of MCFL compared with previous federated multi-modal settings. Further, the success of FedCola shows that *modality collaboration can not only be achieved by feature-sharing between multi-modal data but also can be achieved by parameter-sharing between uni-modal data.*

## 6.3 RESOURCE REQUIREMENTS ANALYSIS

A major benefit of FedCola is its ability to improve modality collaboration without incurring extra computation and communication costs, while CreamFL requires them for feature extraction and transmission. Simultaneously, the absence of a public dataset involvement reduces the risk of privacy leakage from gradient inverse attacks (Wu et al., 2023; Jeon et al., 2021) for FedCola.

To better illustrate the resource requirements, we compute the computation and communication costs of all the clients ($N_v = N_l = 16, r = 0.25$) and the server *per round*. We choose CIFAR-100 as the image dataset, AGNEWS as the text dataset, and 500 samples in MS-COCO as the public dataset for CreamFL. The result is shown in Figure 6. While all methods require a similar communication,

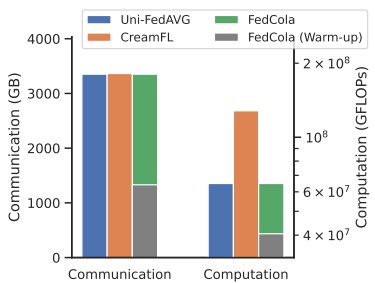

Figure 6: Resource requirements for each method.

Table 5: Impact of each module in FedCola. MAT: modality-agnostic transformer. AS: Attention Sharing. MC: Modality Compensation. MW: Modality Warm-up.

| MAT | AS | MC | MW | Img Acc | Txt Acc | Avg Acc |
|-----|----|----|----|---------|---------|---------|
| ✓ |   |   |   | 3.58 | 88.40 | 45.99 |
| ✓ | ✓ |   |   | 56.17 | 89.67 | 72.92 |
| ✓ | ✓ | ✓ |   | 56.71 | 90.15 | 73.43 |
| ✓ | ✓ | ✓ | ✓ | **57.26** | **90.20** | **73.73** |

Figure 7: One modality's capability vs. the other's performance.

CreamFL requires $1.97\times$ computation. Note that FedCola maintains the same resource requirements as Uni-FedAVG by only optimizing the *strategies* in the key process. Furthermore, when modality warm-up is applied, shown as FedCola (Warm-up), *the resource costs will be significantly reduced due to fewer clients participating* in the *warm-up stage*. Therefore, the efficiency of FedCola further extends the application scope to clients with fewer resources.

## 7 DISCUSSION

**Verification of Modality Collaboration.** To verify our proposed framework provides modality collaboration in MCFL, we further conduct experiments to assess the effectiveness of the modality collaboration. Assuming our proposed framework can facilitate modality collaboration, it is expected that enhancing the model capability for one modality can subsequently impact the performance of the other modality. Therefore, we change the model capability by changing the number of tokens per input, that is, the patch size of the image model and the max token length of the text model, more details are provided in Appendix G. Figure 7 reveals a positive correlation between the increased capability of one modality and the enhanced performance of the other. This evidences that our method indeed possesses the ability to capitalize on out-of-modality knowledge.

**Ablation Study.** As shown in Table 5, the baseline vanilla MAT suffers from a bias towards language, showing an unaccepted vision accuracy. Upon integrating a better parameter-sharing strategy, *i.e.*, Attention Sharing, we observed a noticeable improvement in performance. The image accuracy increases to $56.17\%$, while the text accuracy rises to $89.67\%$, leading to an average accuracy of $72.92\%$. *Note that Attention Sharing has the most significant impact on improving performance*. However, Modality Compensation and Modality Warm-up provide marginal but crucial impacts to further push forward the performance. Specifically, Modality Compensation provides a correction to the misalignment during aggregation, indicating a marginal but crucial improvement to further push the average accuracy to $73.43\%$. To further verify the impact of Modality Compensation, we further perform experiments on scenarios where the numbers of vision and language clients are imbalanced in Appendix D.2. *It shows that Modality Compensation can significantly improve performance beyond Attention Sharing under such imbalanced scenarios* by improving the average accuracy from $71.41\%$ to $73.01\%$. Similarly, Modality Warm-up slightly increases the final performance to $73.73\%$. Furthermore, *it provides additional impacts by reducing resource costs,* as shown in Figure 6 and Section 6.3. All the strategies composite our proposed baseline framework for MCFL, and we hope our baseline can enlighten more studies on modality collaboration.

## 8 CONCLUSION

In this paper, we introduced modality-collaborated federated learning as a novel setting in federated learning. This setting enables modality collaboration, allowing each client to leverage out-modality knowledge to enhance the performance of the global model in its own modality without requiring multi-modal training data. To address the challenges of this new setting, we have proposed Fed-Cola as a robust baseline. FedCola efficiently handles model heterogeneity and bridges the modality gap, leading to significantly improved performance. Comprehensive evaluations have further underscored the effectiveness and efficiency of our proposed method. The advancements presented in this paper open up new avenues for the deployment of FL in multi-modal data scenarios, providing a strong foundation for future work in this promising area.

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

APPENDIX

# A RELATED WORK

**Uni-modal Federated Learning:** Federated Learning (FL) has emerged as a powerful approach for decentralized, privacy-preserving machine learning, where the performance often relates to the amount of available training data (Li et al., 2020b; Mansour et al., 2020). Formally, consider the aggregation process on the server for a parameter $w$ from $N$ client models $w_i$, with each client having $m_i$ training samples. The update of $w$ at round $t$ using the FedAVG algorithm can be expressed as:

$$w^{(t)} = \sum_{i=1}^{K} p_i w_i^{(t)},$$

where $p_i = \frac{m_i}{m}$ and $m = \sum_i m_i$. Here, $m_i$ represents the number of training samples on client $i$.

Given that both the global and client models start with the same initialization $w^{(t-1)}$, the updates on client $i$ can be formulated as

$$w_i^{(t)} = w^{(t-1)} + \eta \nabla F_i(w^{(t-1)}),$$

using gradient descent for $E$ epochs with a learning rate $\eta$ and the local objective $F_i$.

The aggregation can then be further expressed as:

$$\begin{aligned}
w^{(t)} &= \sum_{i=1}^{K} p_i w_i^{(t)} \\
&= \sum_{i=1}^{K} p_i \left[ w^{(t-1)} + \eta \nabla F_i(w^{(t-1)}) \right] \\
&= w^{(t-1)} + \eta \sum_{i=1}^{K} p_i \nabla F_i(w^{(t-1)}),
\end{aligned}$$

with

$$\nabla F(w^{(t-1)}) = \sum_{i=1}^{K} p_i \nabla F_i(w_i^{(t-1)}).$$

In this way, an equivalent back-propagation is achieved without direct accessing the raw data on the clients. While most existing FL research focuses on uni-modal data scenarios with the perspectives of local regularization (Mendieta et al., 2022; Li et al., 2020a), aggregation (McMahan et al., 2017), and initialization (Chen et al., 2022), the challenge to tackle is the *data heterogeneity within one modality*. Furthermore, we identify an opportunity to leverage cross-modal collaboration beyond the uni-modal scenarios and propose our modality-collaborated federated learning (MCFL) setting. Although data heterogeneity is still a challenge in MCFL, the focus of MCFL lies in the model heterogeneity when encoding multi-modal data and the modality gap, which is not shown in previous uni-modal settings. Therefore, we choose the widely-used algorithm FedAVG (McMahan et al., 2017) for the default optimization in our study, so all the methods are using FedAVG for aggregation. However, our study and framework are orthogonal and compatible with other aggregation algorithms such as FedProx (Li et al., 2020a).

**Previous Multi-Modal Federated Learning:** Most previous multi-modal federated learning focuses on scenarios where each entity has multiple representations in different modalities. Therefore, these data are represented in different modalities with aligned semantics. In some settings, each pair of modalities is located in the same client (Xiong et al., 2022; Feng et al., 2023), while in some other settings, each pair of modalities can be split into different uni-modal clients (Yang et al., 2022). Clients with uni-modal data can also assist the multi-modal learning in some settings (Yu et al., 2023), but the motivation to include those clients is only to help the alignment of multi-modal data. However, even uni-modal tasks in different modalities can still share high-level knowledge, which doesn't require the participation of multi-modal data. To completely exclude the reliance on multi-modal data, MCFL is proposed to build a collaboration between uni-modal clients, where no

Table 6: Performance of each strategy.

| Strategy | Img Acc | Txt Acc | Avg Acc |
|---|---|---|---|
| Uni-FedAVG | 51.39 | 88.13 | 69.76 |
| Vanilla MAT | 3.58 | 88.40 | 45.99 |
| Balanced MAT | 5.15 | 88.11 | 46.64 |

additional requirements on these uni-modal clients. Meanwhile, considering the uni-modal clients have their original tasks, the motivation for such a collaboration is to improve the performance of their own tasks with out-modality knowledge. Among all the previous studies, only a line of work, that leverages knowledge distillation (KD), can be adapted to MCFL since they can address the model heterogeneity challenge with the help of a public dataset (Yu et al., 2023; He et al., 2020; Cheng et al., 2021). Further, CreamFL (Yu et al., 2023) achieves state-of-the-art performance in all KD-based methods. Therefore, we choose CreamFL as a competitive comparison method in MCFL.

## B    EXPERIMENTAL SETTINGS FOR EMPIRICAL STUDIES

We choose a general setting shown in Section 6.1 as our setting for empirical studies. Specifically, we choose CIFAR-100 and AGNEWS as the vision and language datasets. For the federated learning settings, each dataset is partitioned into $4$ clients following a Dirichlet distribution with a $\alpha = 0.5$. We will randomly sample half the clients among all clients in each round for communication. The model architecture is described in Section 6.1

## C    DISCUSSION OF THE BIAS IN VANILLA MODALITY-AGNOSTIC TRANSFORMER

Since the default aggregation of FedAVG will weigh the vision clients less due to the smaller number of training samples, we consider a simple fix to balance the aggregation between the modalities with the imbalanced number of training samples. Specifically, we will aggregate the models with the same modalities and then *average* the models' weights from all modalities. We note this simple fix as Balanced MAT. In this way, there is no bias towards any modality at the aggregation level. However, although there is a slight improvement, such a balanced aggregation can still not mitigate the model bias towards language, as shown in Table 6. It indicates that such a bias is not just at the aggregation level but at the learning level. Therefore, our proposed MCFL setting cannot be trivially solved by MAT, highlighting the significance of our proposed research questions.

## D    ANALYSIS OF MODALITY COMPENSATION

### D.1    GENERALIZABILITY ANALYSIS OF FEDAVG AND MODALITY COMPENSATION

In this section, we further show the model aggregated with modality compensation will have the same level of layer-wise alignment as FedAVG. Specifically, the essence of our modality compensation lies not in the overarching generalizability of the entire model but, more specifically, in the *alignment of generalizability at each layer*. Here's a clearer elucidation:

We start from a premise: *If the entire model undergoes aggregation using standard FedAVG, then there should be an inherent uniformity in generalizability across **layers***.

Now, examining the modality compensation in light of FedAVG can offer clarity. For a vision-specific parameter aggregated using modality compensation, the associated weights in the global model $\boldsymbol{w}_g$ can be illustrated as follows:

$$\phi^{(v)<t>} = \sum_{i=1}^{N_v} \frac{|\mathcal{D}_i|}{|\mathcal{D}|} \phi_i^{(v)<t>} + \sum_{i=1}^{N_l} \frac{|\mathcal{D}_i|}{|\mathcal{D}|} \phi^{(v)<t-1>}$$

---

**Algorithm 1** FedCola

---

**Input:** Datasets $D_1, D_2, ..., D_n$ from $n = \sum_{i=1}^{|M|} N_m$ clients with modality $m_i$, number of communication rounds $T$, warm-up rounds $T_w$, warming $m_w$, heat-distribution rounds $T_h$, participating client number $K$

**Initialize:** Global model $\phi^{<0>}$ with modality-specific parameters $\phi^{(m)<0>}$ for each $m$ and shared parameters $\phi^{(m_s)<0>}$.

**for** $t = 1$ to $T$ **do**

    Select $K$ clients from **any** modality;

    **if** $t <= T_w$ **then**

        Further select clients with $m_w$ from the selected clients;        ▷ Modality Warm-up

    **end if**

    ▷ Client Operations

    **for** each selected client $i$ **do**

        **if** $T_w < t <= T_h$ and $m_i \neq m_w$ **then**

            Freeze $\phi^{(m_s)}$        ▷ Heat Distribution

        **end if**

        Client $i$ downloads local model $\phi^{(m_i)<t-1>} \cup \phi^{(m_s)<t-1>}$;

        Client $i$ performs local training using its dataset $D_i$;

        Client $i$ computes new local model $\phi_i^{(m_i)<t>} \cup \phi_i^{(m_s)<t>}$;

    **end for**

    ▷ Server Operations

    **for** each selected client $i$ **do**

        **for** each modality $m \neq m_i$ **do**

            $\phi_i^{(m)<t>} \leftarrow \phi^{(m)<t-1>}$;        ▷ Modality Compensation

        **end for**

    **end for**

    **for** each modality $m$ **do**

        $\phi^{(m)<t>} \leftarrow \sum_{i=1}^{n} \frac{|D_i|}{|D|} \phi_i^{(m)<t>}$;        ▷ Server Aggregation for each modality

    **end for**

    $\phi^{(m_s)<t>} \leftarrow \sum_{i=1}^{n} \frac{|D_i|}{|D|} \phi_i^{(m_s)<t>}$;        ▷ Server Aggregation for shared parameters

**end for**

---

In an ideal scenario, disregarding communication constraints, if we directly send all parameters (as opposed to just modality-related ones) in a multi-modal global model to clients and then aggregate the complete updated models with FedAVG, the aggregation for the vision-specific parameter at the round $t$ would look like:

$$\phi^{(v)<t>} = \sum_{i=1}^{N} \frac{|\mathcal{D}_i|}{|\mathcal{D}|} \phi_i^{(v)<t>} = \sum_{i=1}^{N_v} \frac{|\mathcal{D}_i|}{|\mathcal{D}|} \phi_i^{(v)<t>} + \sum_{i=1}^{N_l} \frac{|\mathcal{D}_i|}{|\mathcal{D}|} \phi_i^{(v)<t>}$$

Given that the language parameters, when sent to image clients, remain untrained, their weights remain unchanged. Thus,

$$\forall i \in [N_l], \phi_i^{(v)<t>} = \phi^{(v)<t-1>}$$

Upon aggregation, these parameters (retaining their values from the last iteration) join the mix, mirroring the action of our modality compensation. Consequently, the outcome of our modality compensation closely aligns with a FedAVG aggregation, fostering *uniform generalizability across layers*.

## D.2 IMPACT OF MODALITY COMPENSATION WHEN CLIENTS NUMBER ARE IMBALANCED

Since the motivation of the modality compensation is to fix the misalignment during aggregation, we assume the impact of modality compensation will be larger when facing a more balanced scenario. We further discover another setting to change the imbalance between modalities. Compared with the original scenario where language samples are more than vision samples, more language clients

($N_l = 16$) will further increase the level of imbalance. On the opposite, we also evaluate a setting that reduces such imbalance, where there are more vision clients ($N_v = 16$). As shown in Table 7, the impact of modality compensation positively correlates with the imbalance.

## E    MORE RESULTS UNDER IMBALANCED NUMBERS OF CLIENTS

The number of clients for each modality will impact the ratio between different modalities during the client selection. To further discuss its impact under imbalanced numbers of clients, we conduct experiments with $\alpha = 0.1$ and $r = 0.25$ under various numbers of clients for each modality. Specifically, we study cases when there are more clients on vision ($N_v = 16, N_l = 4$) and more clients on language ($N_v = 4, N_l = 16$). The results in Table 8 demonstrate the robustness of FedCola in handling imbalances in the number of clients across different modalities. In both the CIFAR-100+AGNEWS and OrganAMNIST+MTSamples settings, FedCola consistently outperforms Uni-FedAVG and CreamFL, indicating its superior capability in adapting to imbalanced multi-modal federated learning environments.

## F    ALGORITHM OF FEDCOLA

In order to show a detailed whole picture of our proposed framework, FedCola, we further present the pseudo-code for the client training and server aggregation in Algorithm 1. We denote the parameters with $\phi_i^{(m)<t>}$, where $i$ means the index of the client parameters, while no subscript means the global parameters, $(m)$ indicates the modality of the parameters, while $(m_s)$ indicates the shared parameters for all modalities, and $<t>$ indicates the round.

## G    IMPLEMENTATION DETAILS

### G.1    INTRODUCTION OF DATASETS

**CIFAR-100 (Krizhevsky et al., 2009):** The CIFAR-100 dataset is a set of 60,000 color nature images distributed across 100 classes, with 50,000 images for training and 10,000 for testing. The dataset is widely used for image classification tasks.

**AGNEWS (Zhang et al., 2015):** The AGNEWS dataset is a collection of web news articles categorized into four classes, with a total of $120,000$ training samples. It is commonly employed for text classification tasks in natural language processing.

**OrganAMNIST (Yang et al., 2023):** OrganAMNIST comprises $58,850$ grayscale images of different organs for medical image classification.

**MTSamples (South et al., 2012):** MTSamples is a collection of $5,000$ transcribed medical reports across various specialties. It is widely used in the field of natural language processing, particularly for tasks related to medical text analysis.

### G.2    VISUALIZATION OF THE DATA PARTITIONING

To give a more intuitive understanding of the heterogeneity in our setting, we provide a visualization of the client data for the datasets in our settings. The visualizations are shown in Figure 8.

Table 7: Impact of modality compensation under unbalanced client numbers

| Schemes | Default $(N_v = 4, N_l = 4)$ | Less Imbalance $(N_v = 16, N_l = 4)$ | More Imbalance $(N_v = 4, N_l = 16)$ |
|---|---|---|---|
| Attention Sharing | 72.92 | 67.83 | 71.41 |
| + Modality Compensation | 73.43(+0.51%) | 67.86(+0.03%) | 73.01(+**1.6**%) |

Table 8: Evaluation results under imbalanced numbers of clients

| Method | CIFAR-100+AGNEWS | | OrganAMNIST+MTSamples | |
|---|---|---|---|---|
| | $N_v = 16, N_l = 4$ | $N_v = 4, N_l = 16$ | $N_v = 16, N_l = 4$ | $N_v = 4, N_l = 16$ |
| Uni-FedAVG | 49.02 | 43.51 | 44.57 | 55.75 |
| CreamFL | 48.24 | 40.55 | 45.70 | 60.71 |
| FedCola | 53.44 | 44.84 | 52.08 | 62.40 |

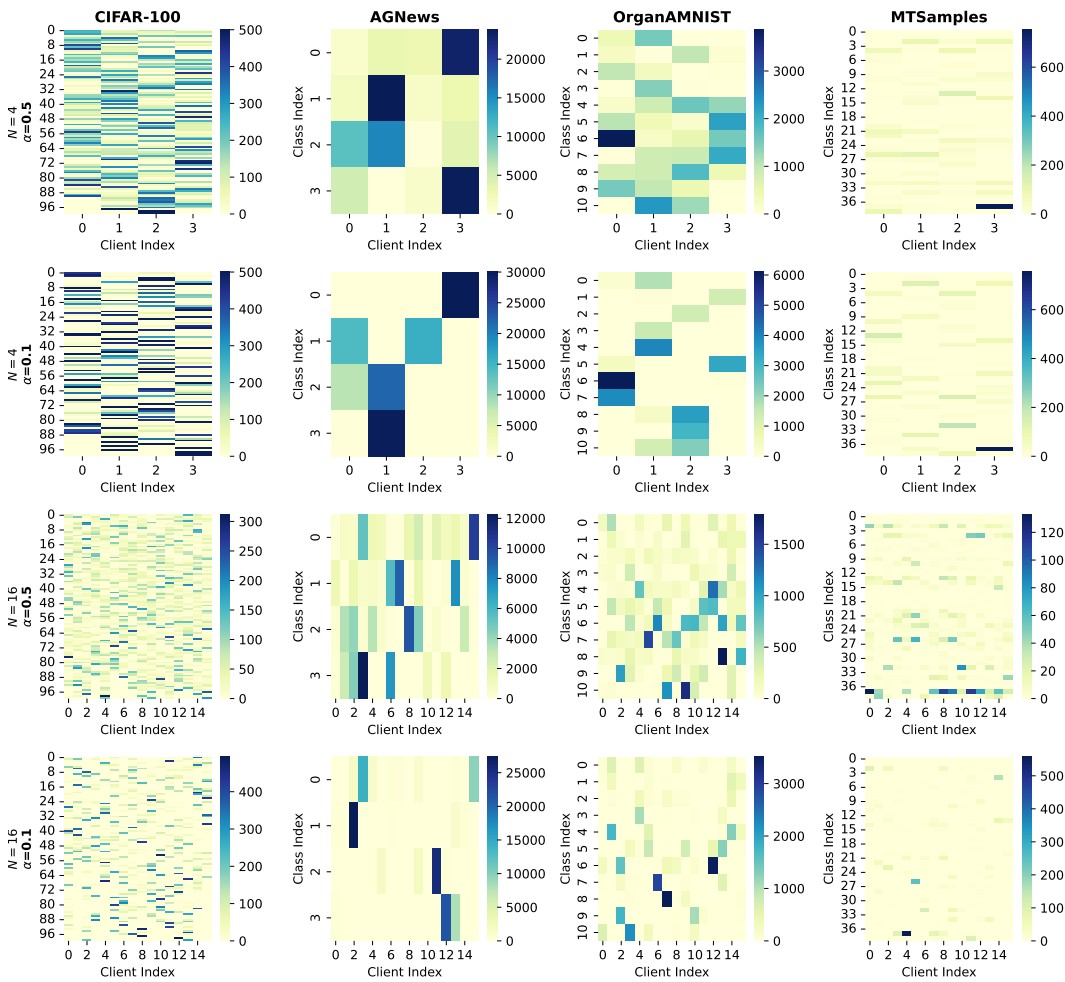

Figure 8: Data partitioning for different settings and datasets

### G.3 METHOD IMPLEMENTATION DETAILS AND HYPERPARAMETERS

To ensure the reproductivity of our experiments, the implementation details are provided in this subsection.

**General Hyperparameters:** We set the local training learning rate for all clients in all methods as 0.0005, and use AdamW (Loshchilov & Hutter, 2017) as the optimizer. The batch size is set as 64 in most cases. The text token length is set as 40 for the language datasets; that is, samples with a shorter token number will be padded to the maximum text token length, while ones with a longer token number will be truncated. All the implementations are based on Pytorch (Paszke et al., 2019), and all experiments are performed using 4 Nvidia A5000 GPUs.

**CreamFL (Yu et al., 2023):** We adapt CreamFL to fit our modality-collaborated federated learning (MCFL) task instead of the original *federated multi-modal learning* task. Since no multi-modal data exists on clients, simply aggregating knowledge with the public dataset will lead to unacceptable performance. We add a standard aggregation with FedAVG besides the multi-modal learning on the server in the original settings to make it a stronger comparison method in MCFL. We set the knowledge distillation weight on the server as $0.01$ with $5,00$ samples in MS-COCO as the public dataset and the inter-intra contrastive weight as $0.01$.

**FedCola:** We set the warming-up and frozen rounds as $5$ for the modality warm-up. The results reported in the main results are with the optimal strategy discussed in Section 5.3. A balanced aggregation scheme is applied by multiplying a scale factor during the aggregation to make the number of samples for each modality equal.

**Verification of Modality Collaboration:** Since a change of the token number is required, we modify the model architecture in experiments and randomly initialize the weights from scratch in a smaller model. Specifically, we employ a 7-layer transformer with the input image size as $32$. To change the token number, we adjust the patch size from $8$ to $2$. Thus, the token number is $16, 64$, and $256$ accordingly.

## H   BROADER IMPACT AND LIMITATIONS

**Broader Impact.** Our framework enables modality collaboration without reliance on multi-modal data. Particularly impactful in scenarios like healthcare, it enables more effective use of sensitive data in various modalities, such as medical images or genomic data, enhancing diagnostic models and patient care. By extending the eligible clients from uni-modality to multi-modality, FedCola could attract more participants to federated learning, enriching shared model diversity and quality.

**Limitations.** FedCola currently does not address system heterogeneity, which represents a limitation in the present framework. We propose to explore this aspect in future research.

