# OpenReview forum: "Exploring Modality Collaboration with Modality-Agnostic Transformers in Multi-Modal Federated Learning"
_ICLR.cc/2024/Conference — Submitted to ICLR 2024_

### Official Review · Reviewer_stt3 · 2023-10-28

**Soundness:** 3 good
**Presentation:** 3 good
**Contribution:** 4 excellent
**Rating:** 8
**Confidence:** 4

**Summary:**

The paper addresses the limitations of current federated learning (FL) frameworks by introducing a novel setting called Modality-Collaborated Federated Learning (MCFL) that focuses on collaboration among uni-modal clients with different data modalities. MCFL aims to leverage the shared knowledge among uni-modal clients while ensuring performance gains across individual modalities, making it a practical and appealing approach for scenarios with diverse uni-modal data. The proposed framework, FedCola, addresses the challenges of model heterogeneity and modality gaps through strategies such as modality-agnostic transformers, attention sharing, and modality compensation.

**Strengths:**

- The paper addresses a significant issue in federated learning and proposes a novel solution.
- Showcases an optimal combination of parameters and strategies to enhance performance.
- The well-designed evaluation is provided across multiple scenarios to affirm the efficacy of the proposed solution.

**Weaknesses:**

Major:

- The proposed framework is based on FedAVG, can other model aggregation methods be applied to further improve the performance?
- Although the proposed framework requires fewer resources than other methods when transformers are applied, what about the resources compared with CNNs?
- During the warm-up stage, are participating clients sampled from only one modality? If so, the comparison might be unfair since more clients on the warm-up modality are participating.

Minor:

- Can this framework handle clients with multi-modal data? For the pointed-out healthcare scenario, one client with multi-modal data is also common.
- Security and privacy are not discussed. Considering the application scenarios (i.e., hospitals), security and privacy are highlighted.

**Questions:**

See the weaknesses

---

> ### Author Response · Authors · 2023-11-20
> **Official Response to Reviewer stt3**
>
> We sincerely appreciate the effort and attention you've devoted to reviewing our paper. Your thoughtful feedback and the high rating have been incredibly helpful and encouraging.
>
> ---
>
> > The proposed framework is based on FedAVG, can other model aggregation methods be applied to further improve the performance?
>
> As highlighted in Appendix A, while our FedCola framework is indeed grounded in the FedAVG algorithm, its essence extends beyond a mere aggregation method. FedCola encompasses a system-level approach, incorporating elements such as model architecture, parameter-sharing strategy, and temporal modality arrangement. Importantly, it is designed to be adaptable and compatible with a variety of model aggregation methods, not limited to FedAVG. This flexibility allows for potential enhancements in performance through the application of alternative aggregation techniques.
>
> > ... what about the resources compared with CNNs?
>
> In the experiments conducted for our study, we employed a ViT-Small model as the backbone, which comprises approximately 21 million parameters. This is comparable to the parameter size of a ResNet50, which contains around 24 million parameters. Therefore, the resource utilization between these two models is similar. However, it's important to note that Convolutional Neural Networks (CNNs) are not an ideal choice for our Modality Collaborated Federated Learning (MCFL) framework. This is because CNNs do not encode multi-modal information as effectively as transformers, which is a crucial capability for the objectives of MCFL.
>
> > During the warm-up stage, are participating clients sampled from only one modality?...
>
> During the warm-up stage of our experiments, we ensured fairness by sampling clients uniformly across **all modalities**, employing a **fixed random seed** for all methods. It's important to clarify that while clients from all modalities are considered during the sampling process, communication during this stage is restricted to those with the specified warm-up modality. This approach means that the number of clients actively participating in communication is actually reduced in the warm-up stage.
>
> As a result of this selective communication with fewer clients, the communication costs associated with the warm-up stage, as depicted in Figure 6, are lower compared to other stages. This reduced cost is a direct consequence of the limited client engagement during this initial phase.
>
> > Can this framework handle clients with multi-modal data? ...
>
> Indeed, our framework is designed to be adaptable and can certainly accommodate scenarios where clients possess multi-modal data. While our primary focus in Modality Collaborated Federated Learning (MCFL) is on a **more challenging** setting where each client has access to only one modality, the framework is fully **compatible** with situations involving clients with multi-modal data.
>
> In such instances, our approach involves sending modality-specific parameters for all pertinent modalities, in addition to the shared parameters, to the client for local training. This enables effective data-level collaboration across different modalities. This setup is particularly relevant and practical in scenarios like the healthcare sector, where it's common for a single client to handle multi-modal data. Our framework's flexibility in this aspect ensures its applicability and effectiveness in a diverse range of real-world settings.
>
> > Security and privacy are not discussed...
>
> You're absolutely right in emphasizing the importance of security and privacy in federated learning, particularly within sensitive domains such as healthcare. In our paper, while the primary focus has been on the technical development of the Modality Collaborated Federated Learning (MCFL) framework, we acknowledge that security and privacy are critical considerations.
>
> It's important to note that the implementation of our MCFL framework does not introduce any new risks beyond those already present in uni-modality federated learning. As such, the existing methods and best practices for ensuring security and privacy in traditional federated learning environments are applicable and effective in the context of MCFL.
>
> We recognize the significance of addressing these issues explicitly and thoroughly. Therefore, we plan to expand upon security and privacy measures in future iterations of our work, ensuring a comprehensive and robust approach to these critical aspects.
>
> ---
> We deeply appreciate your encouraging rating for our paper and are grateful for the support and guidance your feedback has provided. Your constructive comments have played a crucial role in helping us refine and improve various aspects of our work, enhancing its clarity, depth, and overall relevance. If you have further questions, we would like to provide a further explanation. Thank you for the invaluable insights and the confidence you have shown in our research.

---

> ### Author Response · Authors · 2023-11-22
>
> As the interactive rebuttal window will close soon, we sincerely thank you for your valuable feedback. We hope that we have comprehensively addressed the concerns in our response and appreciate any additional feedback you may have. We appreciate your encouraging rating of our paper. Thank you again for your time.

---

> > ### Comment · Reviewer_stt3 · 2023-11-23
> > **My reply to the rebuttal**
> >
> > I appreciate the authors' efforts to address the questions regarding the details of the methodology. It addresses my concerns sufficiently.

---

### Official Review · Reviewer_2xgy · 2023-10-31

**Soundness:** 2 fair
**Presentation:** 3 good
**Contribution:** 2 fair
**Rating:** 3
**Confidence:** 4

**Summary:**

This paper focuses on the research problem of uni-modal clients with different modalities in federated learning. It explores several strategies, including cross-modal parameter sharing, model aggregation, and temporal modality arrangement. The authors provide empirical results to discuss their statement and compare the performance with the selected baseline CreamFL.

**Strengths:**

1. The writing of this paper is easy to follow, and the logic is clear.
2. The main perspectives in Sec. 5 are sound in the multi-modal federated learning.
3. The authors provide extensive experiments.

**Weaknesses:**

1. I have concerns about the motivation of this work. In this paper, each client has one single modality, which is not practical in the real-world setting. Though the authors use the hospital as an example, it is more practical that different clients may have different ratios of different modalities of data.
2. Also, if I understand correctly, as stated in Sec. 5.2, the authors expect a better-aligned global model. However, assuming each client has one single modality, it should fall into the personalized federated learning domain, where we care more about the clients’ local performance.
3. Section 5 is one of the key parts of their proposed work. However, some of the parts are put in the appendix.
4. About the model compensation, I am concerned about the extra communication cost and practicality that we require all the clients’ models to have all the parameters.
5. I am concerned if the core of the technique is still based on the power of transformers which are able to handle different modalities of data.

**Questions:**

Please refer to the weaknesses.

---

> ### Author Response · Authors · 2023-11-20
> **Official Response to Reviewer 2xgy (1/2)**
>
> We sincerely appreciate your thorough review and valuable insights, which have helped us identify areas in our research that require further clarification.
>
>
> > I have concerns about the motivation of this work. In this paper, each client has one single modality, which is not practical in the real-world setting. Though the authors use the hospital as an example, it is more practical that different clients may have different ratios of different modalities of data.
>
> We appreciate your concerns regarding the practicality of our work. However, we believe that our setting is indeed **practical** and highly relevant in numerous scenarios involving **sensitive data**, particularly in the healthcare sector, as Reviwer RJk2 acknowledges our setting *"more realistically reflects the nature of real-world data"*. In a hospital environment, for example, different departments (such as MRI and CT scan units) collect data in various modalities. Due to strict protocols, these departments often **cannot freely share data**, leading to a scenario where each department essentially acts as an independent client with uni-modal data within a federated learning system. This not only demonstrates the practicality of our approach but also highlights its **necessity** in certain contexts.
>
> Moreover, we would like to clarify that our framework is designed to accommodate the scenario where each client has a single modality, but it is **not restricted to** this scenario alone. The modality-specific parameters in our framework can indeed encompass more than one modality. In cases where clients have data in different modalities, we can send the modality-specific parameters for all relevant modalities, along with shared parameters, for local training. The rest of the process remains the same, ensuring **compatibility** with the scenario you mentioned.
>
> Additionally, our focus on scenarios where each client has access to only one modality is intentional, stemming from the **challenge** and its **relevance** to federated learning. When clients possess data in multiple modalities, direct data-level knowledge sharing is feasible, allowing the client model to be trained directly on multiple modalities. In contrast, when a client has access to only single-modality data, knowledge sharing must occur at the parameter level through the aggregation of parameters from other clients. This represents a more complex and **challenging** scenario.
> Meanwhile, it's worth noting that multi-modal learning within a single client aligns more closely with centralized learning paradigms. Our research, however, focuses on cross-modal learning through **parameter sharing without direct data access**, adhering to the core principles of federated learning.
>
> > Also, if I understand correctly, as stated in Sec. 5.2, the authors expect a better-aligned global model. However, assuming each client has one single modality, it should fall into the personalized federated learning domain, where we care more about the clients’ local performance.
>
> We would like to clarify that personalized federated learning and traditional federated learning represent two distinct approaches with differing emphases. Personalized federated learning prioritizes the **local performance** of individual clients, whereas traditional federated learning is more concerned with the **performance of the global model**. Our work focuses on the global performance on each modality, which falls into the latter.
>
> In the context of our work, while it's true that each client possesses data in a single modality, such a modality is **not unique** on one client but shared among multiple clients.
> Our primary objective isn't to further personalize the global model for each specific modality per client.
> Such personalization, if pursued, would be limited to **a single modality**, diverging from the central theme of our paper, which is **modality collaboration**. Our focus remains on enhancing the global model's performance through collaborative learning across different modalities, a concept that aligns with the overarching goals of traditional federated learning.

---

> > ### Author Response · Authors · 2023-11-20
> > **Official Response to Reviewer 2xgy (2/2)**
> >
> > > Section 5 is one of the key parts of their proposed work. However, some of the parts are put in the appendix.
> >
> > The placement of some sections in the appendix was a decision driven by the constraints on the length of the paper. We aimed to include all essential concepts, methodologies, and results in the main paper, while supplementary analyses and extended discussions were moved to the appendix.
> >
> > Specifically, within Section 5, and more precisely in Section 5.2, two parts have been put into the appendix. These sections in the appendix are intended to supplement and enrich the content of the main paper without hindering the understanding of our proposed modality compensation scheme. To illustrate, in Section 5.2 of the main paper, we discuss the motivation and methodology of the modality compensation scheme. The corresponding sections in the appendix, Appendix E and D, provide a detailed algorithm and further proof to enhance reproducibility and facilitate a comprehensive comparison of our scheme with FedAVG.
> >
> > We would be open to suggestions on which specific parts you feel are crucial and should be included in the main text for clarity and completeness in the revised version.
> >
> > > About the model compensation, I am concerned about the extra communication cost and practicality that we require all the clients’ models to have all the parameters.
> >
> > We appreciate your concern regarding the communication cost and practicality of our model compensation approach. However, there seems to be a misunderstanding. As detailed in Section 5.2 and illustrated in Figure 5, the compensation mechanism operates **exclusively at the server level**, thereby **not imposing additional communication burdens** on the clients. This design choice is critical in ensuring that our approach remains both practical and efficient in terms of communication overhead. For further clarity and a comprehensive understanding of our modality compensation mechanism, we encourage a review of the detailed algorithm presented in Appendix E.
> >
> > > I am concerned if the core of the technique is still based on the power of transformers which are able to handle different modalities of data.
> >
> > While it's true that the use of transformers is the starting point of our proposed framework due to its multi-modal capability, it's crucial to understand that our framework extends **beyond merely leveraging a model architecture**. Transformers' success in processing diverse types of data modalities inspired our approach, but our work is a comprehensive solution tailored for federated learning environments, not simply applying a model architecture.
> >
> > As we explore in Section 4, while transformers inherently possess the ability to manage different modalities, simply implementing a transformer model does not automatically address the challenges specific to federated learning.
> >
> > Furthermore, as discussed in Section 5 ("Towards Better Modality Collaboration"), our framework is guided by three critical research questions: 1) Which parameters should be shared among different modalities? 2) How can client models from diverse modalities be effectively aggregated? 3) What is the optimal approach for temporal arrangement in multi-modal learning? The resulting outcome is the complete framework for our proposed novel MCFL setting.
> >
> > We also want to highlight that the exploration of these questions is a significant part of our contribution. Our findings and insights lay the groundwork for future research in this area, demonstrating that the core of our technique is not solely based on the power of transformers but also on the innovative application of these models within the unique constraints and opportunities of federated learning.
> >
> > We hope that our responses have provided further clarity and addressed your concerns about our research. We're grateful for the opportunity to engage in this constructive dialogue and hope that our clarifications meet your expectations. If you have further questions, we would like to provide a further explanation. If you find our revisions and explanations satisfactory, we would kindly request you to consider improving the rating of our paper. Thank you for your time and invaluable feedback.

---

> ### Author Response · Authors · 2023-11-22
>
> As the interactive rebuttal window will close soon, we sincerely thank you for your valuable feedback. We hope that we have comprehensively addressed the concerns in our response and appreciate any additional feedback you may have. We kindly request you to consider improving the rating of our paper. Thank you again for your time.

---

### Official Review · Reviewer_RJk2 · 2023-10-31

**Soundness:** 3 good
**Presentation:** 3 good
**Contribution:** 3 good
**Rating:** 5
**Confidence:** 3

**Summary:**

The paper introduces a novel setting in Federated Learning (FL), termed Modality-Collaborated Federated Learning (MCFL), which focuses on collaboration among uni-modal clients with different data modalities. A new framework termed Federated Modality Collaboration (FedCola), which leverages a modality-agnostic transformer, is proposed to address the challenges in MCFL. Several strategies were probed to optimize the parameter-sharing, aggregation, and temporal modality arrangement in the FedCola. Empirical studies were conducted with two modalities, vision and language, and FedCola showed promising performance for both.

**Strengths:**

- The research investigates an under-explored area in FL, moving from uni-modal to multi-modal data, which more realistically reflects the nature of real-world data.
- The paper presents a very thorough investigation leveraging several strategies to optimize the MCFL framework, including parameter-sharing, aggregation, and temporal modality arrangement.
- The proposed framework, FedCola, is practical and adaptable to more intricate FL scenarios, not limited to the two-modal setting that was experimented.
- The authors provided comprehensive experiments and comparisons, demonstrating the superiority of FedCola over other methods in terms of both performance and resource requirements.

**Weaknesses:**

- The experiments are primarily conducted on two-modal settings, and the adaptability of FedCola to more complex scenarios with more modalities was not thoroughly studied. Could the proposed framework be extended to scenarios with more modalities?
- The effectiveness of temporal modality arrangement was linked to the correlation between different modalities. Will the performance be influenced by the semantics of different modalities?
- Limited discussion on privacy issues: The consideration of privacy issues in federated learning is critical; however, the paper did not sufficiently address this issue in the MCFL.

**Questions:**

- The motivation of the modality compensation is based on an equation based on the number of training samples. However, in the figure provided, the misalignment is based on the number of sampled clients. Can the authors further explain the subtle differences here?
- Is the server model the same as the client models? Can the authors explain more about the relationship between the client and server models?

---

> ### Author Response · Authors · 2023-11-20
> **Official Response to Reviewer RJk2**
>
> Thank you for dedicating your time and expertise to review our paper. Your insightful feedback and the high rating are greatly appreciated, as they not only help refine our work but also motivate our continued research efforts. We are truly grateful for your valuable contributions.
>
> ---
>
> > The experiments are primarily conducted on two-modal settings, and the adaptability of FedCola to more complex scenarios with more modalities was not thoroughly studied. Could the proposed framework be extended to scenarios with more modalities?
>
> Our current research primarily centers on vision and language modalities, but it's important to emphasize that our framework is compatible with scenarios encompassing additional modalities, a topic we explore in Section 5.4. Looking ahead, we are actively planning further studies to apply and refine our FedCola framework in more complex multi-modal environments. We are confident that this future work will further showcase FedCola's adaptability and robustness, expanding its applicability in diverse federated learning contexts.
>
> >The effectiveness of temporal modality arrangement was linked to the correlation between different modalities. Will the performance be influenced by the semantics of different modalities?
>
> Indeed, the semantics of different modalities and their correlation play a significant role in the performance of our framework, as highlighted in our study. We delve into the impact of semantic correlation in Table 3 and provide an in-depth discussion on this subject in Section 5.3.
>
> To illustrate, the correlation level between modalities influences the necessity of the heat-distribution stage in our framework. For example, CIFAR-100 and AGNEWS exhibit a relatively low correlation, implying that the shared parameters trained using the 'warming' modality are not readily adaptable to the 'warmed' modality. In such cases, freezing the weights during the heat-distribution stage can decelerate the adaptation process, rendering this stage less effective compared to scenarios involving modalities with higher semantic correlation, like OrganAMNIST and MTSamples. This aspect of our research underscores the importance of considering semantic correlations when applying our framework to different modality pairings.
>
> >Limited discussion on privacy issues: The consideration of privacy issues in federated learning is critical; however, the paper did not sufficiently address this issue in the MCFL.
>
> We appreciate your concern about the treatment of privacy issues in our MCFL setting. We want to emphasize that MCFL does not introduce additional privacy risks beyond what is inherent in traditional federated learning setups. In MCFL, as is customary in standard federated learning, the clients' data remains local, with only the model updates being transmitted for aggregation purposes. This methodology ensures that the privacy of the data is preserved as safe as in traditional federated learning.
>
> Our focus in this paper was primarily on addressing the technical challenges within the MCFL framework. However, we recognize the importance of privacy and aim to explore this in greater depth in future research.
>
>
> > The motivation of the modality compensation is based on an equation based on the number of training samples. However, in the figure provided, the misalignment is based on the number of sampled clients...
>
> Thank you for highlighting this point. We acknowledge that there is an intrinsic relationship between the number of sampled clients and the number of training samples in our model. However, these two factors can be considered equivalent in our context, as clients are sampled uniformly across the board with a fixed ratio. Consequently, an increase in the number of sampled clients directly translates to a greater volume of training samples.
>
> We have also taken this opportunity to correct a typo in the figure to more accurately depict the alignment. This amendment should provide a clearer understanding of how the number of sampled clients and training samples interact and influence the modality compensation in our framework.
>
> > Is the server model the same as the client models?...
>
> As shown in Section 2 (Problem Definition) and Appendix E, the server model and the client models are different. The server model contains modality-specific parameters for **all** modalities ($\phi^{(m)}$) as well as the shared parameters ($\phi^{(m_s)}$), while the client model only contains the modality-specific parameters for **its** modality and the shared parameters. In this way, unrelated parameters don't need to be involved in the communication, which makes FedCola maintain the same level of communication as FedAVG.
>
> ---
> We are grateful for the opportunity to refine our work based on your constructive feedback. Your comments have been instrumental in helping us enhance the clarity, depth, and relevance of our research. If you have further questions, we would like to provide a further explanation.

---

> ### Author Response · Authors · 2023-11-22
>
> As the interactive rebuttal window will close soon, we sincerely thank you for your valuable feedback. We hope that we have comprehensively addressed the concerns in our response and appreciate any additional feedback you may have. Thank you again for your time.

---

> > ### Comment · Reviewer_RJk2 · 2023-11-23
> > **Response to rebuttal**
> >
> > Thanks for your detailed response. I appreciate the contributions to the community with a new setting. I have no follow-up questions.

---

### Official Review · Reviewer_iEDo · 2023-10-31

**Soundness:** 2 fair
**Presentation:** 2 fair
**Contribution:** 2 fair
**Rating:** 3
**Confidence:** 4

**Summary:**

This submission discusses a scheme where in the multi-modal setting, modality-agnostic transformer models used in different modalities can share parameters of self-attention layers. Authors further suggest parameters specific to individual modality that are not shared can still be augmented to to the cross-modal models and averaging can be taken for the augmented model (referred to as “cross-modal aggregation). Together with warming-up training for each individual modality, authors suggest a framework named FedCola to do federated learning across image-text modalities. Experimental results on CIFAR100 and AGNEWS datasets are reported.

**Strengths:**

The topic of this work is highly relevant to the theme of ICLR.


I very much appreciate authors’ effort to make the narrative to be direct and concise, and to formulate several questions in the text that outline key points in the proposed methodology.

**Weaknesses:**

Several technical details should be further clarified to let the paper to be more convincing.

**Questions:**

- on methodology: what is the frequency of doing inter-modality update of the shared parameters and that of doing intra-modality update of modality-specific parameters?
- on methodology: as the number of training data points/clients in different modalities could be different, the shared parameters should be far more frequently updated compared to modality-specific parameters. It seems that authors have proposed the aggregation step to address this problem. It remains unclear to me how gradient back-propagation works with respect to those parameters which are manually aggregated/augmented into a model, and how these external parameters function during the inference process of a certain modality?
- on methodology: under the assumption of data homogeneity and absence of Byzantine workers, should the Uni-FedAVG method show better performance on the dataset CIFAR100? i.e., for the vision modality itself, I am expecting that the accuracy should be somehow higher than that reported in Table 4. [Benchmarking FedAvg and FedCurv for Image Classification Tasks, Casella et al., 2023]
- on results: regarding reported average accuracy results in tables 1 and 3, how are the average computed? Is there any weighting assigned to each modalities?
- on results: intuitively, when comparing the balance of clients for different modalities, why not show the accuracy change in each modality under different $(N_v, N_l)$ setup?
- on methodology: Convolutional networks are perhaps a simpler type of models for vision related tasks. As in the CIFAR100 case, transformer-based models seem to be somehow below par of the performance of convolutional networks, I wonder if it is possible to incorporate convolutional networks in the study and see relevant results.

---

> ### Author Response · Authors · 2023-11-20
> **Official Response to Reviewer iEDo (1/3)**
>
> Before delving into the specifics of your queries, we would like to extend our sincere gratitude for the time and effort you have invested in reviewing our submission.
>
> ---
>
> > - on methodology: what is the frequency of doing inter-modality update of the shared parameters and that of doing intra-modality update of modality-specific parameters?
> > - on methodology: as the number of training data points/clients in different modalities could be different, the shared parameters should be far more frequently updated compared to modality-specific parameters. It seems that authors have proposed the aggregation step to address this problem. It remains unclear to me how gradient back-propagation works with respect to those parameters which are manually aggregated/augmented into a model, ...
>
> Your questions are raised due to confusion regarding the underlying federated learning (FL) process of our setting. To clarify, the frequency of updates for all parameters, whether they are shared or modality-specific, is **once per round**, so the shared parameters are not more frequently updated.
>  Such updating in most federated learning frameworks is **not directly via back-propagation** but via an aggregation of all client models. However, this process can be conceptually understood as an equivalent operation to performing back-propagation on the global model at the server level.
> Let us provide a formal demonstration of how back-propagation equivalently operates for the global model in FL, and we've added it to Appendix A in the revised version (colored orange):
>
> Consider the aggregation process on the server for a parameter $w $ from $N $ client models $w_i $, with each client having $m_i $ training samples. The update of $w $ at round $t $ using the FedAVG algorithm can be expressed as:
> $$
>   w^{(t)} = \sum_{i=1}^K p_i w_i^{(t)},
> $$
> where $p_i = \frac{m_i}{m}$ and $m = \sum_i m_i $. Here, $m_i$ represents the number of training samples on client $i $.
>
> Given that both the global and client models start with the same initialization $w^{(t-1)} $, the updates on client $i $ can be formulated as
> $$
>     w_i^{(t)} = w^{(t-1)} + \eta \nabla F_i(w^{(t-1)}),
> $$
> using gradient descent for $E $ epochs with a learning rate $\eta $ and the local objective $F_i $.
>
> The aggregation can then be further expressed as:
> $$
>     \begin{aligned}
>         w^{(t)} &= \sum_{i=1}^K p_i w_i^{(t)} \\
>                 &= \sum_{i=1}^K p_i \left[ w^{(t-1)} + \eta \nabla F_i(w^{(t-1)}) \right] \\
>                 &= w^{(t-1)} + \eta \sum_{i=1}^K p_i \nabla F_i(w^{(t-1)}),
>     \end{aligned}
> $$
> with
> $$
> \nabla F(w^{(t-1)}) = \sum_{i=1}^K p_i \nabla F_i(w_i^{(t-1)}).
> $$
> Thus, it results in **one step of update per round** for all parameters. The primary distinction between modality-specific and shared parameters lies in the number of clients where the gradient is averaged: *modality-specific parameters receive a weighted average of gradients from clients with the corresponding modality, whereas shared parameters are averaged from all clients*. As we discussed in Section 5.2, such a distinction may lead to misaligned generalizability among all layers.
> Furthermore, we address this distinction on total client numbers, **instead of frequency**, with our proposed modality compensation scheme, ensuring that both shared and modality-specific parameters are updated effectively to suit their respective roles in the federated learning process.
>
> >..., and how these external parameters function during the inference process of a certain modality?
>
> For the inference process of a certain modality, only the **modality-specific parameters** for that modality as well as the **shared parameters** will be involved in the inference process, whereas the other parameters for external modalities won't be involved.

---

> > ### Author Response · Authors · 2023-11-20
> > **Official Response to Reviewer iEDo (2/3)**
> >
> > > on methodology: under the assumption of data homogeneity and absence of Byzantine workers, should the Uni-FedAVG method show better performance on the dataset CIFAR100? i.e., for the vision modality itself, I am expecting that the accuracy should be somehow higher than that reported in Table 4. [Benchmarking FedAvg and FedCurv for Image Classification Tasks, Casella et al., 2023]
> >
> > In response to your query regarding the Uni-FedAVG method and its anticipated performance on the CIFAR-100 dataset, it is crucial to clarify a few key aspects of our experimental setup. Firstly, our experiments do not operate under the assumption of data homogeneity. Instead, we have adopted a **heterogeneous**, non-IID (independent and identically distributed) setting (Dirichlet distribution), which is typical in the federated learning community.
> >
> > Furthermore, it is important to note that the results referenced in Casella et al., 2023, pertain to **CIFAR-10**, not CIFAR-100. *This distinction is significant, as CIFAR-10 and CIFAR-100 differ in complexity, which can substantially impact model performance*.
> >
> > Additionally, the overall performance of the Uni-FedAVG method is influenced by a range of factors, including the number of clients (10 in Casella et al. vs. 16 in ours), client participation rate at each round (1 in Casella et al. vs. 0.5 and 0.25 in ours), the number of local epochs (1,10 and 30 in Casella et al. vs. 5 in ours), and the specific optimizer settings used. In our methodology, we have deliberately chosen to employ a straightforward approach, avoiding extensive data augmentation or advanced training techniques, while *random horizontal flips* and *angle rotation* are used in Casella et al. This decision is driven by our intention to maintain a clear focus on the primary research questions of our study, allowing for a fair and unbiased comparison across different methods.
> >
> > > on results: regarding reported average accuracy results in tables 1 and 3, how are the average computed? Is there any weighting assigned to each modalities?
> >
> > In Tables 1 and 3, the average accuracy is calculated as an **equal-weighted arithmetic mean**. This method was chosen to treat each modality equally, without introducing bias towards any specific modality. We've modified the description in Section 2 to make it more specific (colored orange).
> >
> > >on results: intuitively, when comparing the balance of clients for different modalities, why not show the accuracy change in each modality under different $(N_v, N_l)$?
> >
> > Thanks for pointing it out.
> > Our primary interest lies in the imbalance in the **total number of training samples**, specifically, CIFAR-100+AGNEWS as a text-heavy setting and OrganAMNIST+MTSamples as an image-heavy setting. However, we agree that the imbalance in the number of clients will slightly influence the client selection process, and we've added the following results under imbalanced settings to Appendix E in the revised version (colored orange):
> >
> > |Method| CIFAR-100+AGNEWS (16, 4) | CIFAR-100+AGNEWS (4, 16) | OrganAMNIST+MTSamples (16, 4) | OrganAMNIST+MTSamples (4, 16)|
> > |-|-|-|-|-|
> > |Uni-FedAVG|49.02|43.51|44.57|55.75|
> > |CreamFL|48.24|40.55|45.70|60.71|
> > |**FedCola**|**53.44**|**44.84**|**52.08**|**62.40**|
> >
> > In both the CIFAR-100+AGNEWS and OrganAMNIST+MTSamples settings, FedCola consistently outperforms Uni-FedAVG and CreamFL, indicating its superior capability and robustness in adapting to imbalanced multi-modal federated learning environments.

---

> ### Author Response · Authors · 2023-11-20
> **Official Response to Reviewer iEDo (3/3)**
>
> >on methodology: Convolutional networks are perhaps a simpler type of models for vision related tasks. As in the CIFAR100 case, transformer-based models seem to be somehow below par of the performance of convolutional networks, I wonder if it is possible to incorporate convolutional networks in the study and see relevant results.
>
> We respectfully disagree with the notion that transformer-based models are generally outperformed by convolutional networks in vision tasks. In recent years, transformer architectures have emerged as a leading choice in various vision-related applications, including image classification[1], object detection[2], and semantic segmentation[3]. Their ability to effectively process and interpret visual data has been well-documented in numerous studies. As in the CIFAR-100 case, Astroformer [4], which is in a transformer architecture, is the current state-of-the-art method according to the benchmark for image classification on CIFAR-100 *without extra training data* on [paperswithcode](https://paperswithcode.com/sota/image-classification-on-cifar-100).
>
> Moreover, when it comes to **multi-modal tasks**, particularly those involving vision-language integration, transformers have become almost the **de facto architecture**[5,6]. This preference is largely due to transformers' superior capability in handling long-distance dependencies, a feature that is particularly valuable in text data and is not as effectively managed by convolutional networks.
>
> Given that our research is primarily focused on multi-modal data within the context of federated learning, we have chosen to leverage the strengths of transformer models. Their adeptness at handling multi-modal data makes them particularly suitable for our goal of facilitating modality collaboration in a federated learning setting. As such, our study concentrates on the use of transformer models, and incorporating convolutional networks into MCFL falls outside the current scope of our research.
>
> [1] An Image is Worth 16x16 Words: Transformers for Image Recognition at Scale, 2020
>
> [2] DETRs with Collaborative Hybrid Assignments Training, 2023
>
> [3] Image as a Foreign Language: BEiT Pretraining for All Vision and Vision-Language Tasks, 2022
>
> [4] Astroformer: More Data Might not be all you need for Classification, 2023
>
> [5] BLIP-2: Bootstrapping Language-Image Pre-training with Frozen Image Encoders and Large Language Models 2023
>
> [6] VLMo: Unified Vision-Language Pre-Training with Mixture-of-Modality-Experts, 2021
>
> ---
> We hope that these responses will address your concerns and offer additional clarity regarding our research. We appreciate the opportunity to engage in this constructive dialogue and would be grateful if you could consider improving the rating of our paper in light of these clarifications. If you have further questions, we would like to provide a further explanation. Thank you for your time and thoughtful feedback.

---

> ### Author Response · Authors · 2023-11-22
>
> As the interactive rebuttal window will close soon, we sincerely thank you for your valuable feedback. We hope that we have comprehensively addressed the concerns in our response and appreciate any additional feedback you may have. We kindly request you to consider improving the rating of our paper. Thank you again for your time.

---

> ### Comment · Reviewer_iEDo · 2023-11-23
> **Keep score as rejection after reading rebuttal**
>
> Authors rebuttal response did not address my main concerns on the methodology as well as the conclusiveness of presented numerical evidence. Based on authors’ rebuttal reply, I find the reported results more dubious. Therefore, I will keep my score as rejection.
>
> (1)	on uni-modal baseline accuracy:
> More clarify is needed about the method Vanilla modality-agnostic transformer and its implementation. The reported benchmark in table 1 that sharing all weights destructs the functionality for the image application in entirety and meanwhile slightly boosts the performance in the text application is highly counter-intuitive and suspicious, as the image and text dataset are utterly irrelevant. In multi-modality learning, in order to benefit one domain by leveraging information from the other domain, the information needs to be well-coordinated. See reference [1,2,3,4,5,6].
>
> Given that the two datasets (CIFAR100 and AG_News; MedMNIST v2 (using the name in the quoted reference in Yang et al., 2023) and MTSamples) used in the experiments are irrelevant, it does not make sense that a network leveraging information from another domain is going to perform better than the model trained in the uni-modal setting with proper finetuning.
>
> [1] A Deep Multi-Modal CNN for Multi-Instance Multi-Label Image Classification, Song et al, 2018
>
> [2] Multimodal deep networks for text and image-based document classification, Audebert et al., 2019
>
> [3] Pixel-BERT: Aligning Image Pixels with Text by Deep Multi-Modal Transformers, Huang et al., 2020
>
> [4] Matching Images and Text with Multi-modal Tensor Fusion and Re-ranking, Wang et al., 2019
>
> [5] Multi-modal Summarization for Asynchronous Collection of Text, Image, Audio and Video, Li et al., 2020
>
> [6] Dynamic Modality Interaction Modeling for Image-Text Retrieval, Qu et al., 2021
>
>
>
> (2)	on averaging of shared parameters:
> Authors make it very clear that there is no explicit gradient-based update that transpires during the weight sharing process (quote Rebuttal 1/3). This has erased any consideration of potential regularization effect in the optimization process, and makes it even more unlikely to let completely irrelevant information from one domain to benefit another.
>
> (3)	on accuracy characterization regarding each modality:
> It does not make sense to compute mean across modalities, when the task in each domain is completely irrelevant to the other. To characterize optimization performance with multiple objectives, conventional characterization methods such as Pareto front [7] should be used.
>
> [7] Multiobjective Optimization: Interactive and Evolutionary Approaches, Lecture Notes in Computer Science, Zitzler et al., 2008
>
> [8] Convex Optimization, Vandenberghe and Boyd, 2004
>
> (4)	on disproportionate data size across modalities and different number of agents across modalities
>
> a.	Quoting authors’ reply (2/3), the result on the dataset combination (MedMNIST and MTSamples) is far from what was reported in table 4 (image classification accuracy greater than 90%).
>
> b.	The AGNews has twice as much data compared to CIFAR100. When the number of clients in the data-abundant text domain is also larger than its counterpart (16 to 4), by the design of the aggregation in (3), the attention weights from the image  domain with less data points will become irrelevant after aggregation, and the resulted performance should thus gravitate towards the scenario reported in table 1 (88.13% text classification accuracy in non-sharing). This contradicts authors’ reply (2/3).
>
> I thank authors’ effort of presenting detailed response, and acknowledge authors’ point that attaining over 50% accuracy on CIFAR100 is of reasonable performances within large transformer models, FedAvg training algorithm and Dirichlet-distributed training data across clients.

---

> > ### Author Response · Authors · 2023-11-23
> > **Thanks for your reply.**
> >
> > Thanks for your reply. We would like to give further clarification on your concerns.
> >
> > ---
> >
> > > (1) on uni-modal baseline accuracy
> >
> > - We would like to clarify that CIFAR-100 and AGNEWS are **not irrelevant** datasets. This combination is applied by previous works on both federated learning [r1] and centralized learning [r2]. It shows that the information in these two datasets can benefit each other.
> >
> > - For the **results in Table 1**, such results are not due to the irrelevance between datasets, but the **imbalance** between different modalities, as discussed in Section 4 and Appendix C. Due to the imbalance in the aggregation, the global model is mostly from the text models, which makes the aggregated global model similar to the aggregated text models.
> > Consequently, the training on the image clients will suffer from the modality gap between the text-dominated global model and local image data.
> > Meanwhile, the training on the text clients will end up being similar to standard uni-modal training, leading to a similar performance as Uni-FedAVG. The reported results are explainable and motivate the exploration of our proposed framework.
> >
> > - **Multi-modal Learning.** We agree that the multi-modal information needs to be well-coordinated, which is the *collaboration* in our proposed setting. When under a centralized setting, the mentioned methods[1-6] are feasible since data in any modality can be directly accessed. However, data cannot be directly accessed in federated learning. Therefore, previous works are infeasible. The **motivation** for our exploration in Section 5 is to enable collaboration between modalities without directly accessing the data. Specifically, we explore the parameter-sharing strategy, cross-modality aggregation scheme and temporal modality arrangement and propose our framework, FedCola.
> >
> > - **Implementation of the Vanilla Modality-agnostic Transformer:** We use a standard multi-modal transformer design, where each modality has its separate embedding layers to encode the input into embeddings and head for classification. The aggregation of the MAT is shown in Figure 2. The embedding layers and head are aggregated among clients with the same modality, and the *entire* transformer blocks are aggregated among all clients. *All codes will be released*.
> >
> > [r1] Multimodal Federated Learning via Contrastive Representation Ensemble, ICLR 2023.
> >
> > [r2] Adaptive Weight Assignment Scheme For Multi-task Learning, IJAI 2018.
> >
> > >(2) on averaging of shared parameters
> >
> > We would like to clarify that the statement *"there is no explicit gradient-based update"* is incorrect. It doesn't mean there is no gradient-based update, but such updates are performed on the client and aggregated on the server. It is a normal process in **all** federated learning frameworks, which is outside the scope of our paper.
> >
> >
> > Furthermore, our framework is **not a regularization-based method**. Our focus is to enhance modality collaboration in federated learning with 1) a better parameter-sharing strategy, 2) a better aggregation scheme, and 3) a better temporal modality arrangement, as discussed in Section 5. As replied to Question (1), such collaboration is not from irrelevant information.
> >
> > > (3) on accuracy characterization regarding each modality
> >
> > We want to clarify that the averaged accuracy is a **evaluation metric** instead of the **training objective** and the tasks are **relevant**.
> >
> > We use the averaged accuracy as a supplement to provide a metric for the overall performance, while the performance on each modality is also reported. Since each client can only access its own client data in federated learning, the multiple objectives you mentioned are not applicable here.
> >
> > > (4) on disproportionate data size across modalities and different number of agents across modalities
> >
> > a. We report the averaged accuracy, i.e., (Image Acc + Text Acc)/2, in the rebuttal. Therefore, it is incomparable to the performance on the single modality. Meanwhile, limited by time, we perform the experiments on the most challenging setting, where $\alpha=0.1$ and $r=0.25$ (Appendix E), so the reported value is lower than the results in Table 4.
> >
> > b. We agree with you that in such a case the performance will gravitate towards Uni-FedAVG performance. However, such a comparison should be between the Avg. Accs. In that case, the Avg. Acc of Uni-FedAVG is 43.51, while ours is 44.84, which is close and consistent with your assumption.
> >
> > ---
> >
> > We sincerely hope you will consider improving the rating of our paper. Thanks for your time and effort in light of these clarifications.

---

> ### Comment · Reviewer_iEDo · 2023-11-23
> **(1/2) Keeping score unchanged after reading second rebuttal reply**
>
> (1) Regarding the meaningfulness of the multi-modal setup in this work:
>
> a. The CIFAR100  (Krizhevsky et al., **2009**) and AG_News (Zhang et al., **2015**) datasets are irrelevant, simply by checking their corresponding definitions. Such training setup renders the entire training task to be untrustworthy, particularly the claim in Table 1 that by leveraging information from AG_News, a network that only has 3% accuracy on CIFAR100 can get boosted to 56% accuracy.
>
> With authors’ multiple round of reply, it becomes clearer that the multi-modality setup (training paradigm and result comparison) in this work is incorrect.
>
> As I stated in my previous comments, the dataset used for multi-modality needs to be meaningfully paired/manually annotated, for instance Flickr30k dataset[cite1], Microsoft COCO [cite2], Visual Genome [cite3], SBU Caption dataset [cite4], Conceptual Captions[cite5], and Conceptual 12M[cite6]. All these datasets for multi-modality study are  public and are actually used in the six references that authors cited earlier in their first round rebuttal reply.
>
> [cite1] From image descriptions to visual denotations: New similarity metrics for semantic inference over event descriptions, Young et al., 2014
>
> [cite2] Microsoft COCO: Common Objects in Context, Lin et al., 2015
>
> [cite3] Visual Genome: Connecting Language and Vision Using Crowdsourced Dense Image Annotations, Krishna et al., 2017
>
> [cite4] Im2Text: Describing Images Using 1 Million Captioned Photographs, Ordonez et al, 2011
>
> [cite5] Conceptual Captions: A Cleaned, Hypernymed, Image Alt-text Dataset For Automatic Image Captioning, Sharma et al., 2018
>
> [cite6] Conceptual 12M: Pushing Web-Scale Image-Text Pre-Training To Recognize Long-Tail Visual Concepts, Changpinyo et al., 2021
>
> b. The two references authors mentioned do not support their claim either.
>
> -- In [r1], the used PAIRED image-text multi-modal datasets are Microsoft COCO and Flickr30k, respectively. It has been made clear in the original text that “_we distribute ... CIFAR100 to 10 unimodal image clients, and AGNEWS to 10 unimodal text clients_” and that “_All unimodal algorithms are extended to multimodal scenarios by operating on each modality separately, e.g., FedGEMS performs per-modality local distillation and entropy-based selective aggregation._” This is similar to [cite7] and [cite8] where knowledge distillation is done to extract information within each modality with unlabeled data.
>
> In other words, there is categorically no interaction between information of CIFAR100 and information of AG_News in the work [r1] cited by authors.
>
> Also, in [r1], only result in the image modality is reported.
>
> [cite7] DistilBERT, a distilled version of BERT: smaller, faster, cheaper and lighter, Sanh et al., 2020
>
> [cite8] Ensemble Distillation for Robust Model Fusion in Federated Learning, Lin et al,, 2020
>
> -- Authors of [r2] manually created 5 tasks and each task only concerns one modality. In [r2], every task in each modality is still trained separately and their loss functions in the training process are manually added together. There is no interaction between information of CIFAR100 and information of AG_News.
>
> c. authors have not yet directly addressed how the pairing between MedMNIST and MTSamples are meaningfully generated for the multi-modal study.

---

> ### Comment · Reviewer_iEDo · 2023-11-23
> **(2/2) Keeping score unchanged after reading second rebuttal reply**
>
> (2) on averaging shared parameters:
>
> As I pointed out earlier, there is no regularisation effect of doing such an averaging for the optimisation process. Considering that directly averaging weights give dubious numerical results when the data in two domains are unpaired and irrelevant, it does not make sense for the weight averaging to force a multi-modal benefit.
>
> (3) meaningfulness of the result presentation:
>
> Computing the arithmetic mean of the accuracy results across different modalities does not make sense, as the number of clients, amount of data and training setup in each modality can vary drastically.
>
> Authors should report the accuracy in each modality under various training settings clearly, and this will take more than a major revision to complete.
>
> (4) contradiction between the rebuttal information and what was reported in the main text:
>
> My question about the contradiction in reported results is not answered directly or explicitly (point 4b in previous reply).
>
> The supplementary table in rebuttal (2/3) concerns $\alpha=0.1$ and $r=0.25$, as marked in orange in the revised manuscript.
> This combination did not appear in the main text Table 4 or Table 1, which makes the result lack a proper reference. Authors did not specify results per each modality in the supplementary table in the supplementary table either.
>
> Considering that authors are experimenting with less than 20 clients in total and the amount of data per client is rather sufficient, the uni-modal accuracy result, for text task or for image task, should not vary significantly given the data distribution and ratio of client participation are the same, regardless of total number of clients.
>
> As I pointed out in my previous comment, the result in the supplementary table and the result in the main text are inconsistent.
>
>
>
> In conclusion, there is reasonable doubt about the correctness of the setup of this work and the numerical result reported herein.

---

> > ### Author Response · Authors · 2023-11-23
> > **Thanks again for your reply**
> >
> > (1)
> > a. (i) We **didn't claim** that changing from 3% to 56% on the image dataset is due to the leveraging information from AGNEWS. This is due to a better **parameter-sharing strategy** as discussed in Section 5.1. Leveraging information from AGNEWs will increase from *51%* (Uni-FedAVG) to 56% instead of 3%. (ii) Our setting is a **new**, instead of **incorrect**, setting compared with previous settings, which is demonstrated in **Fig. 1** and discussed in **Section 1** (Page 2). Your statement, *"the dataset used for multi-modality needs to be meaningfully paired/manually annotated"*, is one of the motivations making our setting challenging and practical.
> >
> > b & c. Our underlying principle is that *training with more data can get stronger generalizability*. However, your claim that different datasets cannot be trained together to get better performance is **not a valid assumption**. Under such an assumption, pre-training on a larger dataset cannot improve the performance either due to *irrelevance*. Meanwhile, In our MCFL settings, no correlation required between different modalities is one of the **strengths** of our setting, which shows better feasibility than previous settings.
> >
> > (2) The aggregation (averaging weights) doesn't directly provide a multi-modal benefit but a benefit in terms of **generalization**. It is a similar idea to fine-tuning a pre-trained model. In the local training in each round, the client trains its client more with a more generalized global model to improve the performance, which does not necessarily require explicit relevance.
> >
> > (3) We report the **separate accuracy as well as the averaged accuracy** in every result in the main paper (Table 1,3,4,5).
> >
> > (4)
> > We choose the most challenging setting by reducing both the $\alpha$ and $r$ to demonstrate the robustness. Due to the limited time, we cannot directly provide the results on the settings you refer to now, but we would like to add them to the camera-ready version. However, we directly explain your concern with the results from the same setting (43.51 vs. 44.84). It is unfair to assume the results are incorrect as we will release all the codes and models.
> >
> > ---
> >
> > We sincerely hope our responses have provided further clarity and addressed your concerns about our research.

---

> ### Comment · Reviewer_iEDo · 2023-11-23
> **Keeping score unchanged after reading third rebuttal reply**
>
> (1) The major concern on the main method of this submission is that it does not make sense to manually pull two utterly irrelevant datasets, conduct jointly training, **directly add up model weights and take average** and simply claim that the training results are better because of using more data.
> -  In the federated setting, authors can take a reference with [FedMultimodal:
> A Benchmark For Multimodal Federated Learning, Feng et al., KDD 2023] to explore commonly used multi-modal dataset and present a clear argument about the comparison between the fusion scheme presented therein and the proposed method.
> - Also, [A unified framework for multi-modal federated learning, Xiong et al., Neurocomputing, 2022] is another explicit example where a weight matrix is used to correlate feature from different modalities. There needs to be a **non-trivial action leveraging inter-modality information**.
>
> (2)	Methodology for generalization
> - In my previous reply, I have already given specific examples of multi-modal learning where I expressed the point that inter-modality tasks need to be related.
> - Authors have emphasized on parameter sharing, while throughout the main text there is **no explicit discussion** about the difference between proposed method and some established multi-task learning or feature alignment work. Insufficient discussion with prior related work thus incurs the need for major revision of the current manuscript.
> - Extant methods for improving generalization that are similar to what was suggested in this work are representation learning (domain generalization/adaptation, feature alignment), multi-task learning or transfer learning. In these established directions of handling information across multiple domains/modalities, involved tasks/domains are either information-wise relevant or sharing certain overlap of data. I give examples below to demonstrate **specific measures to leverage inter-domain information**:
>     - Domain generalization:
>        - Kernel method is used to explicitly correlate features from different domains [Blan21]
>        - Multi-domain MMD-based regularization: a Maximum Mean Discrepancy penalty between feature distributions is added to loss for learning feature representations across different domains [Li2018]
>
>     - Multi-task learning:
>        * For hard parameter sharing, Caruana [CAR98] made clear that “_uses the training signals for related tasks … to improve generalization_” and so are the examples (steering directions and traffic condition determination) given therein.
>        * For soft parameter sharing, Duong et al.[Duong15] use Frobenius norm based penalty and Yang et al. [Yang17] use trace norm based penalty to regulate parameters in different tasks.
>
> [Li2018] Domain Generalization with Adversarial Feature Learning, Li et al., 2018 CVPR
>
> [Car98] Multitask Learning, R. Caruana, 1997 Machine Learning Springer
>
> [Duong15] Low Resource Dependency Parsing: Cross-lingual Parameter Sharing in a Neural Network Parser, Duong et al., 2015, ACL
>
> [Yang17] Trace Norm Regularised Deep Multi-Task Learning, Yang and Hospedales, 2017 ICLR
>
> [Blan21] Domain Generalization by Marginal Transfer Learning, Blanchard et al., 2021 JMLR
>
> (3) Inconsistency in reported results:
> - Exploring the varying number of clients and presenting comparable results with content in the main manuscript is a comment in my original comment (dated Oct 31, 2023). Authors had the entire rebuttal period to prepare _comparable_ $(N_l, N_v)$ accuracies with the content in the main text ($\alpha$,$r$ setup), but did not present such an update. Claiming lack of time is an invalid excuse.
>
> (4) Dataset choice for experiments:
>
> Authors fail to justify the selected two pairs of text-image datasets. Both datasets have completely irrelevant text and image components, which violates the current common practice in multi-modal learning.

---

### Official Review · Reviewer_9GDZ · 2023-12-04

**Soundness:** 2 fair
**Presentation:** 2 fair
**Contribution:** 2 fair
**Rating:** 3
**Confidence:** 5

**Summary:**

This paper investigates model aggregation strategies in multi-modal settings, where clients may possess diverse data modalities. To tackle the challenges arising from model and data heterogeneity, the authors introduce the FedCola framework, incorporating the following key elements: 1) aggregation of embedding layers within each modality, and 2) aggregation of only the attention layer in the transformer block across multiple modalities. By employing additional techniques such as modality compensation and warm-up, the authors showcase enhancements in the overall performance of the global model.

**Strengths:**

1.	The paper delves into a novel setting within federated learning, with practical applications highlighted, such as in real-world scenarios like hospital data.
2.	The logical presentation of the paper, addressing challenges before presenting potential solutions, enhances its overall coherence and ease of comprehension.
3.	The authors offer comprehensive details on experimental settings, particularly in the context of heterogeneous data distribution, adding depth to their exploration.

**Weaknesses:**

The following feedback highlights specific areas for improvement, including addressing surprising findings, clarifying counter-intuitive results, completing incomplete sections, providing comparisons with alternative methodologies, and adopting a more uniform writing style.

**Questions:**

1)	It is a bit surprising for the reviewer to see the performance on CIFAR-100 could be significantly boosted (5%-10%) by training with AGNews. Given the general perception of the two datasets as having irrelevant data classes, the paper draws a highly counter-intuitive conclusion.  This paper draws a (very) counter-intuitive conclusion, but does not provide sufficient reasoning.
2)  In table 1, the performance of image acc drops to 3% by model aggregation in a traditional FL way. The authors attribute this decline to data imbalance. Notably, data imbalance also exists in Attention sharing and FFN sharing, yet these exhibit good accuracy levels, presenting a counter-intuitive scenario. The paper lacks sufficient explanations for these observations. Moreover, based on the presented results, aggregating FFN or Attn only maintains a good performance. But why aggregating them together leads to a significant drop?
3)	Section 5 is not complete. The modality compensation in Sec 5.2 is supposed to be more clear, whereas the authors leave the algorithm in appendix. It also remains unknown to me how the warm-up is set in Sec 5.3, e.g., what is the round of warm-up or when should we be confident the warm-up is complete.
4)	In table 3, the warm-up for vision leads to acc improvement on AGNews, but a significant drop on MTSamples. As such, readers are not sure such a warm-up is necessary for vision tasks.
5)	The reviewer suggests that a straightforward approach to federated learning (FL) on multi-modality data would involve performing model aggregation for each modality and subsequently using a classical method for multi-modal information exchange, such as latent vector alignment. The paper is expected to include a comparison of results obtained through this methodology, providing a benchmark for evaluating the proposed approach against a more conventional FL strategy for multi-modal data.
6)	The reviewer strongly recommends that the authors refrain from overemphasizing words or sentences in multiple ways throughout the paper. Instances such as using italic font, underlining, employing colored text (e.g., orange) for phrases like "Equal-weighted arithmetic mean", and using bold font for "Top-1 Acc" after Eq 1 are noted. Adopting a more uniform and straightforward writing style is advised to enhance the overall readability and professional appearance of the paper.

In general, the reviewer believes this paper should undergo a major revision, and is not ready to be accepted based on its current status, especially for a top-tier conference like ICLR.

---

### Official Review · Reviewer_pPNy · 2023-12-04

**Soundness:** 2 fair
**Presentation:** 2 fair
**Contribution:** 2 fair
**Rating:** 5
**Confidence:** 4

**Summary:**

This manuscript aims to address a significant problem in federated learning - enabling multimodal learning collaboration among uni-modal clients. The proposed solution of decomposing the modality-agnostic transformer into embedding layers, the feature extraction transformer layers, and task-specific head is reasonable and promising. The experimental results are also provided to illustrate the effectiveness of the proposed method.

**Strengths:**

It provides a simple but useful framework for multmodal learning collaboration among uni-modal clients.

**Weaknesses:**

However, I have the following three main concerns:
- Different modalities shared the transformer layers, which assumes that the outputs for these layers of different modalities are in the same semantic abstract levels.
- It lacks visualizing the representations of the samples from different modalities are well aligned in the feature spaces.
- The solution should be verified on the widely used multimodal learning datasets, e.g., COCO-Captions.

**Questions:**

How can we guarantee that the feature representations from different shared transformer layers for different modalities are in the same semantic abstract levels?

---

### Official Review · Reviewer_Pvz2 · 2023-12-05

**Soundness:** 2 fair
**Presentation:** 2 fair
**Contribution:** 2 fair
**Rating:** 5
**Confidence:** 4

**Summary:**

The paper presents "Modality-Collaborated Federated Learning (MCFL)," a new approach in Federated Learning (FL) that facilitates collaboration across uni-modal clients with diverse data modalities. It introduces the Federated Modality Collaboration (FedCola) framework, utilizing a modality-agnostic transformer for effective cross-modal parameter sharing and model aggregation. The authors demonstrate FedCola's superiority over existing FL solutions, making it a robust baseline for MCFL in multi-modal data environments.

**Strengths:**

1. The paper is well-written, exhibiting clear and coherent logic that is easy to follow.
2. The authors have introduced a scenario in the field of Federated Learning (FL), specifically how to facilitate collaboration among uni-modal clients with different data modalities. This consideration is crucial for advancing the field and demonstrates a deep understanding of real-world challenges in FL.
3. In the process of proposing their methodology, the authors explore three key questions, each meticulously analyzed to formulate the final Federated Modality Collaboration (FedCola) framework. The logical progression through these questions strengthens the validity of the proposed method.

**Weaknesses:**

1. The paper's applicability is questioned due to the two-modal settings. Federated learning between many different medical centers, each with multi-modal and multi-domain data, presents a more realistic scenario. The author should provide a stronger motivation for the new setting, demonstrating why traditional multi-modality and personalized methods are not suitable, as without this it might be challenging to determine whether the methods are specifically designed for the limited scenario.
2. The reason for combining disparate single-modality datasets like CIFAR-100 and AGNEWS is unclear. The paper does not adequately explore the impact of modality similarity on collaborative learning. This raises the question of whether combining unrelated modalities, such as medical images and arbitrary internet text, would be beneficial for the training.
3. The experimental setup, only including OrganAMINIST and MTSamples, lacks sufficient diversity in multi-domain modalities. A simple and straightforward setup would involve clients with single modality and domain, such as three clients with distinct imaging datasets (NIH Chest X-rays, OrganAMINIST, BraTS) and two with textual datasets (MTSamples, MIMIC-III).
4. The effectiveness of larger models remains unexplored. The choice of a small model (ViT-Small) and datasets limits the study’s relevance to larger-scale, cross-modality scenarios typical in medical institutions. Moreover, the baseline selection is too limited (FedAvg and CreamFL). The inclusion of existing multi-modal alignment strategies is necessary for a more comprehensive evaluation.

**Questions:**

see weakness

---

### Author Response · Authors · 2023-11-20
**General Response to Reviewers**

We would like to extend our sincere gratitude to all the reviewers for their insightful and constructive feedback. It is particularly encouraging to note the positive reception of

(1) our proposed new setting, Modality Collaborated Federated Learning (MCFL):
- `Reviewer RJk2`: *"...investigates an under-explored area in FL..."*
- `Reviewer stt3`: *"...addresses a significant issue in federated learning ..."*

(2) the experiments and effectiveness of our baseline framework, FedCola:
- `Reviewer RJk2`: *"...provided comprehensive experiments and comparisons, demonstrating the superiority of FedCola over other methods in terms of both performance and resource requirements"*
- `Reviewer 2xgy`: *"...provide extensive experiments."*
- `Reviewer stt3`: *"The well-designed evaluation is provided across multiple scenarios to affirm the efficacy of the proposed solution."*

(3) the soundness and clarity of the presentation:
- `Reviewer iEDo`: *"appreciate authors’ effort to make the narrative to be direct and concise, and to formulate several questions in the text that outline key points in the proposed methodology."*
- `Reviewer RJk2`: *"...presents a very thorough investigation..."*
- `Reviewer 2xgy`: *"The writing of this paper is easy to follow, and the logic is clear."*

We are delighted to acknowledge the high ratings and the recognition of our contributions in this emerging field.

In response to the reviews, we have made concerted efforts to address and clarify the primary concerns raised about the clarity of our setting and specific details within our proposed framework. Each concern and question raised by the reviewers has been meticulously addressed in separate responses, ensuring that our methodology, intentions, and results are communicated with greater clarity and precision.

---

### Meta-Review · Area_Chair_VFAE · 2023-12-05

**Metareview:**

This paper proposed the FedCola framework for the Modality-Collaborated Federated Learning scenario, which focuses on collaboration among uni-modal clients with different data modalities. The proposed method includes a modality-agnostic transformer for effective cross-modal parameter sharing, model aggregation and temporal modality arrangement, with results on the CIFAR-100/AGNEWS/OraganAMINIST/MTSamples.

This paper initially received very polarized reviews. We also invited an additional 3 FL experts to be the emergent reviewers and reevaluate this submission. During the review and discussion period, reviewers raised concerns and suggested that the paper should 1) clarify the motivation and experimental settings (collaboration between uncorrelated modalities), and the current settings are not convincing to reviewers; 2) provide additional comparisons with more baselines or alternative methods, and demonstrate the practical effectiveness with more real-world datasets or clients with multimodal datasets; 3) clarify the technical novelty and experimental results; 4) discuss the privacy issue and computation cost of the proposed method.

All reviewers agree that this paper still needs further revision for improvement, and its current form is not ready to be accepted at ICLR. Hence, I recommend “reject” for this paper.

**Justification For Why Not Higher Score:**

During the review and discussion period, reviewers raised concerns and suggested that the paper should 1) clarify the motivation and experimental settings (collaboration between uncorrelated modalities), and the current settings are not convincing to reviewers; 2) provide additional comparisons with more baselines or alternative methods, and demonstrate the practical effectiveness with more real-world datasets or clients with multimodal datasets; 3) clarify the technical novelty and experimental results; 4) discuss the privacy issue and computation cost of the proposed method.

**Justification For Why Not Lower Score:**

NA

---

### Decision · Program_Chairs · 2024-01-16

Reject